# Glacial isostatic adjustment reveals Mars's interior viscosity structure

A. Broquet[1✉], A.-C. Plesa[1], V. Klemann[2], B. C. Root[3], A. Genova[4], M. A. Wieczorek[5], M. Knapmeyer[1], J. C. Andrews-Hanna[6] & D. Breuer[1]

Investigating glacial isostatic adjustment has been the standard method to decipher Earth's interior viscosity structure[1,2], but such an approach has been rarely applied to other planets because of a lack of observational data[3,4]. The north polar cap of Mars is the only millions-of-years-old surface feature that can induce measurable surface deformation on this planet, thereby holding clues to its present-day internal viscosity structure[5,6]. Here we investigate the emplacement of this ice cap by combining thermal evolution models[7], viscoelastic deformation calculations[8] and radar observations[6]. We show that downward motion of the northern regions is ongoing and can be constrained by analyses of the time-variable gravity field[9] and NASA's InSight seismic moment rate[10]. Only models with present-day high viscosities (2–6 × 10²² Pa s for depths greater than 500 km), strong mantle depletion in radiogenic elements (more than 90%) and thick average crusts (thicker than 40 km) are consistent with the negligible flexure beneath the polar cap seen by radars. The northern lithosphere must deform at less than 0.13 mm per year and have a seismic efficiency less than 0.3 to satisfy gravity and seismic constraints, respectively. Our models show that the north polar cap formed over the last 1.7–12.0 Myr and that glacial isostatic adjustment can be further constrained by future gravity recovery missions to Mars[11,12].

The response of a planet to loading is intimately linked to its interior structure[1,2]. On Earth, studying the time-variable response of the lithosphere to the growth and decay of large ice sheets, or glacial isostatic adjustment, has been the standard approach to constrain our planet's upper mantle viscosity structure[1]. However, such models have rarely been applied to other terrestrial bodies due to a lack of observational data[3,4]. Mars harbours two geologically young (less than 100 million years old) and large (more than 1,000 km across) polar ice caps[13], which represent the only millions-of-years-old surface features that can induce measurable viscoelastic deformation on this planet. In the absence of in situ heat flow measurements, the analysis of these deformations is one of the few methods that give access to the present-day thermal state and interior structure of Mars[5–7].

Orbital radar sounders have mapped the Martian ice caps[5,14] and the lack of clear measurable lithospheric flexure beneath the north polar deposits (Fig. 1) was used to constrain the planet's interior to be cold with a thick and stiff elastic lithosphere[5,6]. Whereas the results of these studies are used as constraints on the present-day thermal state of Mars's interior[15,16], geodynamic evolution models[7,17,18] struggle to explain the thick and cold lithosphere inferred at the north pole (Extended Data Fig. 1) and, at the same time, the planet's young volcanism[19,20] and ongoing plume activity[21]. This indicates that the assumptions of elastic flexure in previous models should be revisited to account for the interior's transient viscoelastic response.

Most previous analyses have assumed the polar deformation to be at equilibrium, which is only valid if the time elapsed since the polar cap formed is greater than the time required for viscous adjustments[5,6]. Geologic observations and global climate models suggest the north polar cap is only a few million years old, but the exact age remains uncertain[22,23]. Owing to this young age, calculations suggested that viscoelastic relaxation could result in estimating thinner elastic lithospheres and higher heat flows beneath the north polar cap[5,6]. However, these models assumed a constant mantle viscosity, did not account for the ice loading history and only considered a single wavelength to represent the load. As a result, limited insights into the possible effects of viscoelastic relaxation were provided. Here we investigate glacial isostatic adjustment on Mars in light of newly acquired constraints on the planet's interior structure from NASA's InSight mission[7,15,24–26]. Thermal evolution models[7] are used to parameterize the viscoelastic interior structure and are further constrained by the long-term loading history of the north polar cap and radar observations. The north polar deformations are limited based on the observed planetary time-variable gravity field[9] and the InSight-derived seismic moment rate[10], allowing to provide tight constraints on Mars's interior viscosity structure.

## Mars's interior structure from geodynamic models

Without plate tectonics on Mars, incompatible radioactive heat sources, which were sequestered in the crust by magmatic processes,

[1]Institute for Planetary Research, German Aerospace Center, DLR, Berlin, Germany. [2]GFZ Helmholtz Centre for Geosciences, Potsdam, Germany. [3]Delft University of Technology, Delft, The Netherlands. [4]Sapienza University of Rome, Rome, Italy. [5]Universite Paris Cité, Institut de Physique du Globe de Paris, CNRS, Paris, France. [6]University of Arizona, Tucson, AZ, USA. ✉e-mail: adrien.broquet@dlr.de

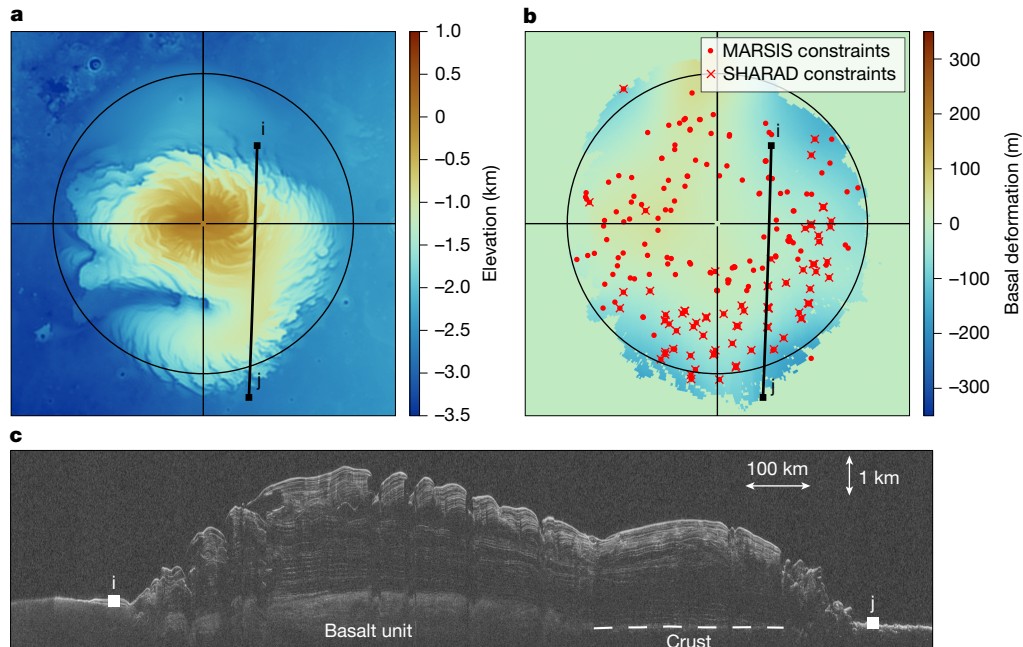

**Fig. 1 | Deformation beneath the north polar cap as seen by radars.**
**a**,**b**, Context image of the north pole (**a**) and interpolated deformed basement assuming a real dielectric constant of 3.0 based on 78 SHARAD and 213 MARSIS radar measurements (**b**) (Methods). **c**, SHARAD depth-corrected radargram (orbit 51924) showing negligible downward flexure. The radargram apparent depth was corrected by assigning real permittivities of 1.0 and 3.0 above and below the detected ground surface. The radar track (i, j) is shown in **a**, **b** and **c**. We note that because of its higher frequency, SHARAD does not penetrate through the sand-rich basal unit beneath the centre of the north polar cap[5].

are unlikely to have been reincorporated into the mantle in any substantial amount[27]. This naturally led to a strong fractionation, in which more than half of the planet's heat producing element content may be present in the crust[28]. In this so-called stagnant lid regime, the crust further acts as a physical boundary that thermally insulates the interior, dictating the rate at which Mars cools in time[7,18]. Thus, the structure and properties of the crust are key to decipher the geodynamic evolution of Mars throughout its geologic history.

We simulate Mars's three-dimensional (3D) thermal evolution and interior viscous flow using the geodynamic code GAIA[29] and including the latest constraints from InSight on the interior structure, namely core size and crustal thickness[25,30] (Methods). Extrapolating InSight local crustal thickness measurements to planetary scale using gravity and topography data is non-unique, and different models accounting for variations in crustal density are tested[26]. Crustal heat production is assumed to be laterally constant, but is ranged from the Gamma Ray Spectrometer measured surface average of 49 pW kg[-1] (ref. 28), to up to twice that value. In total, 84 global geodynamic models are created, among which a subset of models covering a wide range of interior viscosity and temperature are retained for further analysis (Methods, Extended Data Table 1 and Extended Data Fig. 2).

The present-day interior structure beneath the north polar cap is represented for each model by averaging all cells poleward of 60° N in the geodynamic models. In all simulations, the interior shows a stratified viscosity structure with a thick stagnant lid (thicker than 350 km) on top of a lower viscosity mantle ($10^{21}$–$10^{24}$ Pa s) with upper mantle (100–400 km) thermal gradients of 2–3 K km[-1], consistent with earlier work[16,31,32]. For reference, our subset of interior structures are plotted alongside pre-InSight models[18] (Fig. 2). Models with a lower or higher viscosity would require crustal thicknesses thinner or larger than constrained by InSight[26], respectively. In all models, a larger crustal thickness and a higher concentration of heat producing elements in the crust result in a cooler and stiffer mantle. In absence of additional constraints, all selected models are equally possible.

## Ice history and viscoelastic relaxation

Global climate models considering Mars's recent orbital history[33] suggest that the current north polar cap formed over the past 4 million years, with ice-sheet thickness increasing in response to a gradual decline in polar insolation[23]. A geologically recent formation is also supported by crater counting statistics and stratigraphic relationships that indicate ages of only a few million years, albeit with significant uncertainties[22]. Although these studies suggest a young age for the current north polar cap, its precise age and formation history remain uncertain[13]. Former ice caps may have also existed[23,34], but the planet's orbital and climate history cannot be uniquely determined beyond the last several tens of millions of years[33], making it difficult to infer longer term ice histories. If the ice loading time is shorter than or comparable to the adjustment time of the lithospheric mantle (tens to hundreds of millions of years), the equilibrium may not be reached resulting in an ongoing viscoelastic downward deflection. This effect would prominently affect all estimates of the strength of the elastic lithosphere and planetary thermal state, in which viscous relaxation is not addressed adequately[5,6,35].

Viscoelastic relaxation is investigated by computing the time-variable and wavelength-dependent load Love number *h′* using the ALMA code, which solves the momentum equation of a spherically symmetric stratified viscoelastic planet[8] (Methods). Our calculations use a Maxwell rheology, as it is established to describe long-term glacial isostatic adjustment processes[1,2]. Because the age and long-term loading history of the north polar cap are poorly known, we assume a linear increase in thickness over time, leading to the current state, with tested final ages ranging from 100 thousand years ago (ka) to 1 billion years ago (Ga). Although nonlinearities in the ice-accumulation history are expected[36], these would have a negligible effect on present-day deformations (Extended Data Fig. 3). The effect of former north polar caps on the current polar deformations is also explored (Methods). Love numbers are computed for each of our interior models up to spherical harmonic degree 50 and at 300 times spaced evenly on a log scale from 100 ka to 1 Ga. Changes in the seasonal cap whose load is three orders

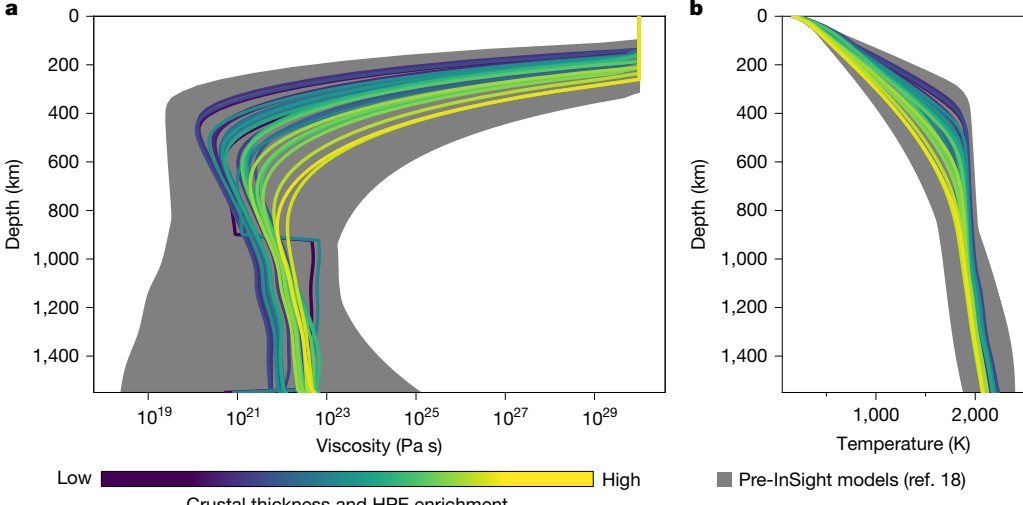

**Fig. 2 | Present-day average interior structure of the northern regions (poleward of 60° N). a,b,** Coloured lines show the viscosity (**a**) and temperature (**b**) structure of the interior models that match InSight constraints. Line colour reflects the average mantle temperature, which is related to the thickness of the crust and its enrichment in heat producing elements (HPE). Detailed model description is shown in Extended Data Fig. 1 and Extended Data Table 1.

of magnitude lower than the perennial cap[37] and that are insensitive to the interior viscosity[4] are not considered.

## Constraints from radar observations

The present-day north polar cap has a thickness of about 3.1 km and bulk density of $1,200 \pm 300$ kg m$^{-3}$, as constrained from radar analyses[6,34], and together with our specified ice loading history these are used to predict the time-variable polar deformations (Methods). For each model, the time-integrated deformations are compared to radar observations of the undeformed northern cap basement from Mars Advanced Radar for Subsurface and Ionosphere Sounding (MARSIS[14]) and SHAllow RADar (SHARAD[5]). The misfit is evaluated using a root-mean-square (r.m.s.) and our viscoelastic and radar analyses consider realistic combinations of permittivity and density for a mixture of dust, water and dry ice[6] (Methods). Accounting for a wide range of uncertainties affecting radar measurements, including instrument range resolution, surface roughness and the uncertainty in the estimation of the preloading polar cap basement, the r.m.s. misfit should be no more than 266 m for MARSIS and 175 m for SHARAD (Methods). For all models, the flexure beneath the polar cap increases as a function of time (or polar cap age) and is associated with an increase in the cumulative radar misfit (sum of SHARAD and MARSIS misfits, Fig. 3a). At all investigated time steps, flexure is greater for lower viscosity interiors due to these models reaching equilibrium more rapidly.

## Mars's time-variable gravity field

Ongoing deformations affect the planetary gravity field[38] and in particular the long-wavelength zonal degrees 2 and 3. For each model, we compute the present-day yearly surface deformation rate linked to mantle flow. In all cases, the younger the polar cap, the higher the present-day yearly deformation rate, which is related to the time-variable viscoelastic response of the interior (Fig. 3b). These values are compared to observational constraints from orbital tracking of the Mars Global Surveyor, Mars Odyssey and Mars Reconnaissance Orbiter missions[9], which show that the planetary degree 2 and 3 zonal gravity field coefficients have marginally increased over 8 Martian years of spacecraft tracking (Extended Data Fig. 4). This variation indicates that the increase in the gravitational potential associated with long-term ice accumulation is higher than the decrease in gravitational potential from downward deflection. Based on climate

models, ice-accumulation rates have been constrained to range from 0.5 to 1.1 mm per year at the north pole[23,39]. Considering density ratios between the accumulating ice and flowing interior mantle, the above minimum accumulation rate implies that ongoing downward deformation in the northern regions should be less than 0.13 mm per year to match the observed time-variable gravity field (Methods).

## Low polar strain rates

If the northern strain rates were too large, crustal failure would occur, potentially leading to the detection of a marsquake by InSight[10]. Therefore, the polar deformations can further be constrained by comparing the geodynamic and seismic moment rates[40], as has been done on Earth[41]. Based on predicted magnitudes from the InSight catalogue, a 3.8 to 3.9 magnitude event originating from the polar regions (75–90° N) could be detectable[10]. Thus, the non-observation of such marsquakes during the InSight mission can be used to invert for the maximum northern lithosphere strain rate (Methods). This approach shows that the viscoelastic strain rate should be less than $1.84 \times 10^{-18}$ s$^{-1}$ to avoid inducing a 3.8 magnitude marsquake poleward of 75° N.

Together, the above constraints are used to limit our multidimensional parameter space encompassing the interior structure, age and bulk properties of the north polar cap. A model is deemed acceptable if the r.m.s. misfit between the present-day deformation and radar measurements is less than both 266 and 175 m for MARSIS and SHARAD, respectively, if the combination of tested density and dielectric constant produce an existing mixture of ice and dust, and if both the deformation and strain rates do not exceed 0.13 mm per year and $1.84 \times 10^{-18}$ s$^{-1}$, respectively. If the polar cap were older, it would have had more time to adjust leading to a vertical displacement below the ice cap larger than observed, whereas if it were younger, deformation rates would be too large and inconsistent with InSight seismic moment rate and Mars's gravity field (Fig. 3a,b). In our inversion, radar measurements provide the tightest constraints on the interior viscosity and limit the maximum age of the polar cap, whereas the time-variable gravity provides a lower bound on the polar cap age. The InSight moment rate establishes a weaker constraint on the polar cap minimum age, but is not in contrast with our gravity analyses.

## Highly viscous mantle and young ice cap

Only three members of our geodynamically based model ensemble were found to be consistent with the small amount of deformation

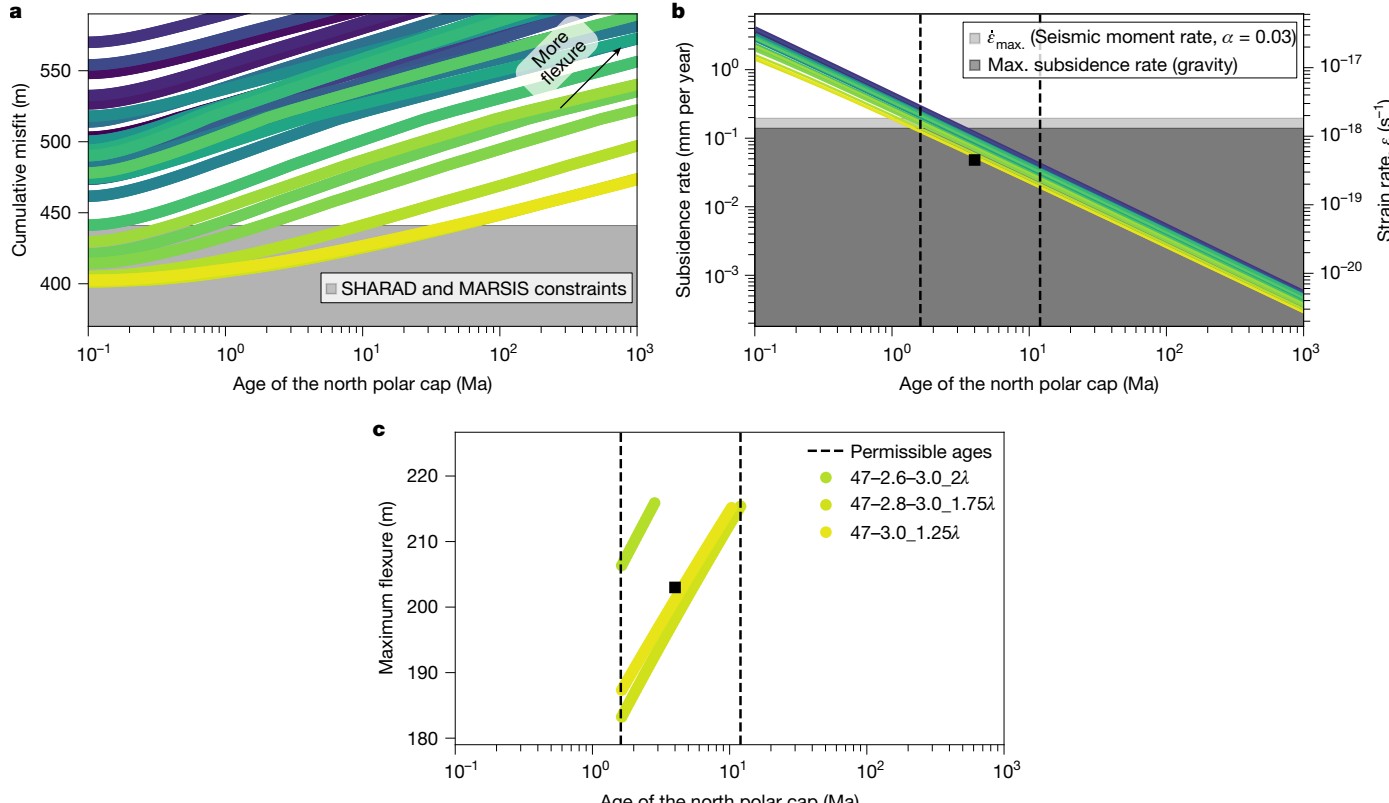

**Fig. 3 | Observational constraints on glacial isostatic adjustment from radar measurements, time-variable gravity and InSight moment rate.**
**a**, North polar cap age versus cumulative misfit of viscoelastic and measured deformation from SHARAD and MARSIS data for all models. **b**, Time-variable deformation and strain rate for all models. **c**, Polar cap age and maximum constrained by radar data, time-variable gravity field and InSight seismicity (Fig. 3c). These models are characterized by a large present-day mantle viscosity with volumetric average values of about $2-6 \times 10^{22}$ Pa s from 500 km down to the core and thick average crust (thicker than 40 km, Figs. 3 and 4). Models with a lower viscosity would predict too much north polar deformation, whereas models with a higher viscosity would require crustal thicknesses larger than measured by InSight[26]. Although our inversion constrains the interior structure of the northern regions, it is comparable to the global average, albeit marginally colder (Extended Data Fig. 5). In our accepted models, the mantle is highly depleted in radiogenic elements compared to the crust, the latter of which must have a heat production rate of at least 61 pW kg$^{-1}$ for our 69-km average crustal thickness model, and up to 98 pW kg$^{-1}$ for our 43-km crust model. These values are 1.25 and up to two times greater than the surface average[28] and indicates that more than 90% of the planet's heat producing element content is present in the crust. This shows that the lower crust has a heat producing element concentration and composition different from that of the surface, as previously suggested[35,42,43]. In agreement with previous work[26,43], our estimated average crustal thickness implies that the crust-mantle interface at the InSight location corresponds to the deepest observed reflector (that is, the three-layered model[30]). Our low preferred mantle temperature further agrees with that inferred from the postolivine transition detected at 1,000 km by InSight[44].

Our models limit the age of the north polar cap load to be 1.7–12.0 Ma, which is consistent with independent global climate models suggesting ages of a few million years[23]. If the age of the polar cap or interior structure of Mars were known, our framework would enable to place precise constraints on either one, potentially allowing to refine the long-term orbital history of the planet and ice-accumulation history of the north polar cap. Based on our allowed deformation rate and the

deformation beneath the north polar cap for accepted models. The colours are the same as in Fig. 2 and the black square in **b** and **c** shows values for the 4 million years old polar cap loading model in Fig. 4. Further information on models is given in Extended Data Fig. 1 and Extended Data Table 1. Max., maximum. Ma, million years ago.

non-observation of high-magnitude marsquakes poleward of 75° N, the ratio between the northern lithosphere deformation rate and seismic moment rate (that is, the seismic efficiency) is found to be at most 0.3. This value is more than two times lower than found using InSight data alone[10]. Such a low seismic efficiency helps explain why InSight has detected fewer marsquakes than expected[24] and may be related to large crustal porosity or the presence of volatiles in the crust.

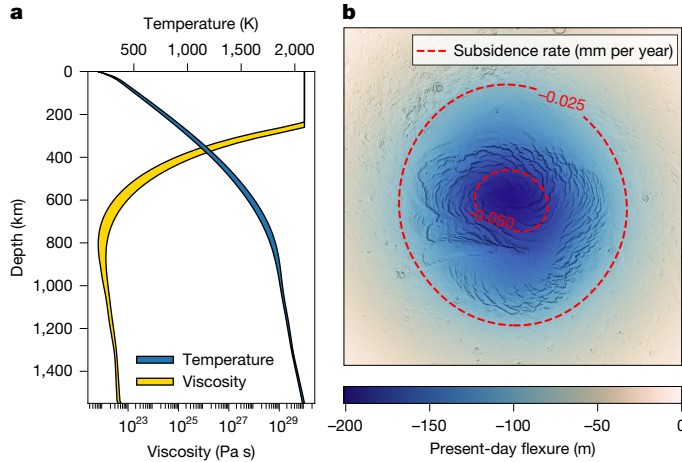

**Fig. 4 | Constrained present-day average interior structure of the northern regions (poleward of 60° N) and polar deformations. a**, Accepted interior viscosity and temperature ranges of the northern regions. **b**, Example of the present-day deformation and subsidence rate for a polar cap age of 4 million years for our 47–3.0_1.25λ model (Extended Data Table 1).

Given the long interior response to loading, we have considered models accounting for the potential existence of former north polar caps. Simulations assuming ice cap formation following periodic obliquity cycles only moderately increased the north polar deformation as a result of continuing adjustment of the ancient uncompensated loads (Extended Data Figs. 6–8). This would not affect our inferred mantle viscosities or heat productions, but would lower the maximum current north polar cap age by a few million years. One way to help reduce the ongoing deformation, and thus our inferred interior viscosity, would require the present-day polar cap to have formed on the sedimentary-infilled flexural trough of a former ice cap. In that framework, the ongoing downward deformation would compete with the rebound related to the past ice cap. However, although such sedimentary infilling may be related to the basal unit beneath the central portion of the polar layered deposits[34], it is not seen beneath the Gemina Lingula region near 80° N (Fig. 1). Therefore, this competing effect alone cannot account for the negligible flexure observed in that region.

Thus far, analyses of glacial isostatic adjustment have been limited to Earth applications because of a lack of observational data. Our work shows that there is opportunity to study this process on another planet and to unveil the interior structure, geodynamic history and long-term orbital evolution of Mars. Furthermore, our analysis sheds light on viscosity variations in the interior of a stagnant lid body, for which long-term geodynamic constraints are scarce. The ongoing subsidence rates of $10^{-2}$ to $10^{-1}$ mm per year predicted by our models affect the time-variable planetary gravity field, a signal that could be directly measured by future space missions to Mars[11,12] thereby bridging the gap in geodynamic modelling and interior exploration between Mars and Earth.

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

## Methods

### Constraints on the deformation beneath the north polar cap

The deformation beneath the north polar cap is obtained by comparing estimates on the polar cap thickness from elevation and radar data following ref. 6. In that work, the polar cap thickness was estimated by subtracting the observed ice cap elevation from a preloading surface constructed using elevation data far from the polar cap and unaffected by flexure (Extended Data Fig. 9). MARSIS radar analyses were performed at 213 locations spatially scattered across the polar deposits. All available MARSIS radargrams close to each location were investigated, and the reflections arising from the icy surface and the ice–substratum interface were visually identified. Using the same framework, analyses of SHARAD radargrams were performed at those 213 locations. Because of its higher frequency, SHARAD does not penetrate through the sand-rich basal unit and we here only retain locations where the ice–crust interface is observed ($n = 78$).

The radar thickness depends on the dielectric properties of the icy materials, which are not uniquely known for the entire north polar cap. On the other hand, viscoelastic deformations depend on both the interior structure and density of the loading materials. Therefore, we compare our simulated viscoelastic deformations to radar measurements using an r.m.s. misfit function for all interior models considering a real dielectric constant range of 2.5 to 3.5 and an ice density of 920 to 1,500 kg m$^{-3}$ covering the possible presence of water and dry ice and dust in the ice cap[5,6]. We limit our ice density and dielectric constant parameter space by considering possible mixtures of dust, water and dry ice using a three-component Maxwell–Garnett mixing model[6]. The density of water and dry ice is considered to be 920 and 1,560 kg m$^{-3}$, respectively, and the dust density is assumed to range from 2,200 to 3,400 kg m$^{-3}$. The dielectric constants of these same materials are 3.0, 2.5 and 6.0, respectively[5,6]. Because of these, a model with an ice density of 920 kg m$^{-3}$ (pure water ice), but with a dielectric constant of 3.5, can be safely ruled out as it does not correspond to any existing mixture of ice and dust in the north polar cap.

By accounting for uncertainties linked to the range resolution of MARSIS, surface roughness at the scale of the Fresnel zone of MARSIS and the uncertainty when estimating the preloading surface, an upper limit on the r.m.s. was estimated to be 266 m (ref. 6). Using the same approach and because of the higher frequency and resolution of SHARAD, a lower maximum allowed misfit of 175 m is given to the SHARAD comparison. Thus, a model is deemed acceptable if the r.m.s. misfit for MARSIS and SHARAD is no more than 266 and 175 m, respectively.

For context, Fig. 1b shows a flexure map beneath the north polar cap that was obtained by interpolating the radar-derived flexure using a 400-km moving window to obtain a smooth basement map and to get rid of short-wavelength uncertainty in elevation and radar-derived polar cap thickness. For comparison, a similar map using a dielectric constant of 2.75 instead of 3.0 is shown in Extended Data Fig. 9. In both cases, the present-day deformation beneath the north polar cap is no more than a few hundreds of metres.

### Thermal evolution modelling

We have run 84 Martian 3D geodynamic models considering a wide range of parameters using the GAIA code[7,18,29]. Therein, conservation equations of mass, linear momentum and thermal energy, are solved from 4.5 Ga to the present day given a set of model parameters. Key parameters that control the present-day thermal state of the interior are core radius, crustal thickness and radiogenic heat production, some of which have been recently constrained by InSight[7,25,30]. Our geodynamic simulations use the 3D structure of the crust, as derived from gravity, topography and InSight data[26]. Absolute viscosities can be obtained using the Arrhenius viscosity law and considering reference values for the viscosity, pressure and temperature[7].

All tested models use the following naming convention: $d_c^{InSight} - \rho_{south}(-\rho_{north})$, where $d_c^{InSight}$ is the crustal thickness at the InSight landing site in kilometres and $\rho_{north}$ and $\rho_{south}$ are the bulk density of the northern and southern hemisphere crust in grams per cubic centimetre (Extended Data Fig. 1 and Extended Data Table 1). If only one crustal density, $\rho$, is indicated, this value is assumed to be constant across the planet. If added, $\lambda$ provides the crustal heat producing element enrichment factor with respect to the nominal Gamma Ray Spectrometer measured average (below). Models with additional annotations have higher activation energy ($E = 325$ kJ mol$^{-1}$), initial temperature ($T_{init} = 1,850$ K), temperature difference across the mantle ($\Delta T = 2,205$ K) and have a $\eta_{jump}$ 25-fold mantle viscosity jump compared to the reference values used in most other cases (that is, $E = 300$ kJ mol$^{-1}$, $T_{init} = 1,650$ K, $\Delta T = 2,000$ K, no viscosity jump, see Extended Data Table 1 for further details). The effect of these parameters within their allowed possible ranges is minor compared to crustal thickness (Fig. 2).

In our nominal models, crustal heat production is laterally constant and equal to the surface average of 49 pW kg$^{-1}$, estimated from Gamma Ray Spectrometer data[28]. However, because the spectrometer instrument is only sensitive to the first upper tens of centimetres of the surface[45], different composition and heat production may exist at greater depth. Therefore, we also consider models in which the bulk crust has a heat production different from the near surface, with an enrichment factor, $\lambda$, of 0.83 and up to 2. The upper bound leads to heat production of 98 pW kg$^{-1}$, which is slightly higher than the highest measured crustal heat production of 75 pW kg$^{-1}$ (ref. 28), making it a reasonable upper range. Because the number of radiogenic elements in a planetary interior is finite, some models with a thick crust cannot reach crustal enrichment factors of 2, as the mantle is already fully depleted. Thus, our approach establishes an upper limit for Mars's mantle viscosity (and the lowest possible temperature) within our geodynamic evolution framework. Whereas higher viscosities may be obtained using a higher reference viscosity, these models would be mostly conductive and be unable to explain ongoing volcanism on Mars.

Geologic observations and geophysical models indicate the presence of melts in the Martian interior[19–21]. All of our models are able to predict melting at the present day when considering the present-day Martian solidus as estimated in ref. 46. From our initial set of 84 geodynamic models, we select a subset of 27 end-member models with a wide range of viscosity structure for our viscoelastic analyses (Extended Data Table 1).

### Thermal profile to lithosphere elastic thickness

A thermal profile can be converted to an elastic lithosphere by setting the bending moment of an elastic plate equal to that of the bending stresses in a more realistic rheology that considers fracturing and viscous flow[47] as

$$\frac{E_y T_e^3 K}{12(1 - v^2)} = \int_0^{T_m} \sigma_{YSE}(z)(z - z_n)dz, \tag{1}$$

where the left term corresponds to the analytic integration of the bending moment of an elastic plate with thickness $T_e$, and in which $v$ is Poisson's ratio, $E_y$ is Young's modulus and $K$ is the lithosphere curvature. The right term computes the bending moment limited by the yielding strength of the lithosphere ($\sigma_{YSE}$), and where $z$ is the depth, $z_n$ the depth of the neutral plane and $T_m$ is the mechanical thickness[47]. The yield strength of the lithosphere strongly depends on temperature variations and includes faulting and viscous stresses. The mechanical thickness is defined as the depth where bounding stresses are close to zero, here approximated to 50 MPa (refs. 5,47).

Thermal profiles from our interior models are converted to an equivalent elastic thickness using the TeHF code[48] that solves the above equation. We consider a wide range of model parameters including, a dry and wet diabase rheology for the crust, a dry and wet olivine rheology

for the mantle. For each rheology, we consider strain rates of $10^{-17}$ to $10^{-20}$ s$^{-1}$ consistent with the timescale of mantle convection and polar cap formation[18] (Fig. 3). The curvature of the lithosphere is also varied from $10^{-9}$ to $10^{-11}$ m$^{-1}$ (ref. 6). For all models, this conversion approach leads to elastic thicknesses less than 270 km, which is inconsistent with constraints from elastic flexure indicating elastic thicknesses more than 330 km (refs. 5,6, Extended Data Fig. 1). This demonstrates that previous elastic modelling must be revisited to consider the interior transient viscoelastic response.

## Calculation of viscoelastic deformation

The vertical displacement at time $t_{end}$, colatitude and longitude ($\theta, \phi$), are determined by the convolution of the spectral representation of the load potential change over the considered time interval with the transfer function represented by the load Love number $h'$ as

$$W(t_{end}, \theta, \phi) = \int_0^{t_{end}} \sum_{l=0}^{l_{max.}} \sum_{m=-l}^{l} \frac{V_{lm}(t)Y_{lm}(\theta, \phi)}{g_0} h_l'(t_{end} - t) dt, \quad (2)$$

where $g_0$ is the gravitational acceleration at the surface, $l$ and $m$ are the degree and angular order, $Y_{lm}$ are real spherical harmonics and $l_{max.}$ is the maximum spherical harmonic degree used for calculations. The change in the load potential of the north polar cap ($V_{lm}$) is calculated assuming a mean bulk density ranging from 920 to 1,500 kg m$^{-3}$ and using the polar cap thickness derived from elevation data and radar measurements[5,6]. For simplicity and because of a lack of observational data, we solely increase the polar cap thickness over time, based on the current morphology of the north polar cap. We do not consider any radial expansion and growth.

Love numbers are computed based on the internal structures predicted by thermal models (Extended Data Fig. 2) using a Maxwell rheology and up to $l_{max.} = 50$, which is largely sufficient to resolve the polar cap loading (roughly degree 8, wavelength of 1,250 km). We show that the effect of using a transient rheology such as Andrade is minor compared to the uncertainty in mantle viscosity structure, and note that it does not change the time behaviour for the considered long-term process (Extended Data Fig. 10). The ALMA code assumes an incompressible interior. Compressibility affects Love number calculation, leading to slightly higher deformations. This effect is considered as second order when compared to the uncertainty in Mars's interior structure.

Each interior model from our thermal evolution simulation is discretized into 68 constant-thickness layers with a given density, viscosity and rigidity, from the core to the surface. This number of layers is chosen to optimize the calculation of the load Love number using ALMA. The interior rigidity is computed using Perple_X (ref. 49) together with the interior temperature, pressure and TAYAK mineralogy[7,26,27].

This study uses average one-dimensional interior profiles to constrain viscoelastic deformations beneath the north polar cap. Although future work may achieve larger accuracy by modelling 3D deformations from a 3D interior structure and polar cap load, several mitigating factors should be noted. Our 3D geodynamic models indicate that Mars's northern regions show homogeneous properties (rigidity, viscosity, temperature) comparable to the global average (Extended Data Fig. 5). This indicates that lateral variations in the interior structure have minor effects on the estimated deformations. In particular, we expect these effects to be markedly lower than the current uncertainty in Mars's interior structure (Fig. 2). Furthermore, the axisymmetric shape of the north polar cap load reduces the influence of the polar cap geometry on deformation estimates. For these reasons, differences between a full 3D viscoelastic loading model and our current model are expected to be small.

## Polar cap loading history and past ice caps

On Mars, polar caps have grown and decayed following the planet's obliquity cycles[23]. It is not uniquely known how obliquity has varied in the planet's recent history (500 million years ago), as orbital models become chaotic as one goes backward in time[33]. Yet, it is generally thought that past polar caps existed in the geologically recent history[23,34], and these may affect the present-day deformation in the northern regions.

To model this process, we have constructed a loading history in which north polar caps grow and decay. Our first model starts at 430 Ma with no polar cap, and builds up a 3.2-km-thick polar cap in 5 Myr following a half cycle of a cosine function, removes that polar cap in 5 Myr and then repeats with a new cycle. The radius of the edge of the polar cap is constant. The interior response associated with the ice cap periodic growth and decay is modelled over the full process duration using the viscoelastic formalism described above. The two end-member viscosity structures from our ensemble of geodynamic models are tested with this time-loading history. In both cases, the flexure values at the present day are 35 and 45 m larger than when neglecting this past history for the low- and high-viscosity models, respectively (Extended Data Fig. 6).

We have also tested these same interior models against the ice-accumulation history of ref. 23. This model starts at 10 Ma with a past north polar cap that disappears at about 8 Ma. After a time interval with no north polar cap, a new polar cap forms with a near linear increase in thickness from 4 Ma to the present day (Extended Data Fig. 7). A key unknown is what was the state of the 10 Ma ice cap and whether it was long-standing and at equilibrium or more recently formed. In one model, we assume that the previous ice cap was present over the last 500 Ma and thus at equilibrium (Extended Data Fig. 7) and in a second model, we assume that it formed at 14 Ma (Extended Data Fig. 8). In the case where the ice cap was long lasting, the present-day deformation increased by up to 70 m, whereas it only increased by a few metres for a young former ice cap.

The above tests show that the deformation beneath the north pole can only be increased when considering the effect of ancient ice caps. Whereas this effect would reduce our inferred maximum age for the polar cap by a few million years ago, it would not affect our constraints on Mars's interior structure.

## Constraints on polar deformation from time-variable gravity

Previous work inverted for Mars's static and time-varying gravity field on the basis of tracking of Mars Global Surveyor, Mars Odyssey and Mars Reconnaissance Orbiter missions over 8 Martian years[9]. Here we build on that work by analysing residual trends in $C_{20}$ and $C_{30}$ after the zonal harmonics are corrected for seasonal variability. In both cases, the sign of these coefficients is negative, indicating the planet's gravitational oblateness and north–south asymmetry. We correct the time-varying signal of these zonal harmonics for annual, semi-annual and tri-annual variations, as well as half (5.5 years) and full solar cycle periods (11 years, ref. 9). The function used to fit the both $C_{20}$ and $C_{30}$ time-variable coefficients is in the form of

$$
\begin{aligned}
f(t) = {} & P_1 \sin\left(\frac{2\pi}{T_1}t\right) + P_2 \cos\left(\frac{2\pi}{T_1}t\right) + P_3 \sin\left(\frac{2\pi}{T_2}t\right) \\
& + P_4 \cos\left(\frac{2\pi}{T_2}t\right) + P_5 \sin\left(\frac{2\pi}{T_3}t\right) + P_6 \cos\left(\frac{2\pi}{T_3}t\right) \\
& + P_7 \sin\left(\frac{2\pi}{T_4}t\right) + P_8 \cos\left(\frac{2\pi}{T_4}t\right) + P_9 \sin\left(\frac{2\pi}{T_5}t\right) \\
& + P_{10} \cos\left(\frac{2\pi}{T_5}t\right),
\end{aligned}
\quad (3)
$$

where $P_1$ to $P_{10}$ are the best-fit periodic coefficients, $t$ is time in seconds past J2000, $T_1 = 365 \times 86{,}400 \times 1.880894$, $T_2 = 365 \times 86{,}400 \times 0.940447$, $T_3 = 365 \times 86{,}400 \times 0.626965$, $T_4 = 11 \times 86{,}400 \times 365$ and $T_5 = 11 \times 86{,}400 \times 365/2$.

After this correction, the residuals show trends with slopes of $1.5 \pm 1.6 \times 10^{-18}$ and $2.5 \pm 3.3 \times 10^{-19}$ s$^{-1}$ for the $C_{20}$ and $C_{30}$ coefficients,

respectively (Extended Data Fig. 4 and Extended Data Table 2). Given the negative sign of $C_{20}$ and $C_{30}$, positive trends suggest polar ice accumulation and that the south polar cap is accumulating less ice compared to the north polar cap. For those time-variable coefficients the Pearson correlation coefficients ($r$ values) are 0.13 and 0.06, respectively. We further test the null hypothesis that the residuals have no correlation with time using Wald's test. For the $C_{20}$ and $C_{30}$ coefficient residuals, we obtain $P$ values of 0.35 and 0.46, respectively, which are both higher than typical significance levels of 0.05. Together, these values indicate a positive slope, although not statistically significant.

After 1 Martian year, our analysis indicates a $\Delta C_{20}$ value of $4.7 \times 10^{-11}$ and $\Delta C_{30}$ of $7.7 \times 10^{-12}$. Together, these can be used to derive the north polar gravitational acceleration as

$$a_m \approx 3\frac{GM}{R^2}\Delta C_{20} + 4\frac{GM}{R^2}\Delta C_{30}, \tag{4}$$

where $G$ is the gravitational constant, $M$ and $R$ are Mars's mass and radius. Based on this equation, our measured $\Delta C_{20}$ and $\Delta C_{30}$ imply north polar gravitational acceleration of $6.4 \times 10^{-5}$ mGal. Using the Bouguer plate formula, $2\pi G\rho h$, and a water ice density of 920 kg m$^{-3}$, the above gravity acceleration suggests a maximum ice-accumulation rate of 1 mm per year when not accounting for isostatic adjustment. This value is consistent with predicted rates of 0.50 to 1.06 mm per year by climate models[23,39], indicating that gravity residuals provide information on north polar processes.

Our models predict that glacial isostatic adjustment is ongoing, with downward deformation rates of $10^{-4}$ to 4 mm per year (Fig. 3). In our framework, glacial isostatic adjustment pushes the mantle downward such that the gravitational signature of this process should be scaled by mantle density (3,500 kg m$^{-3}$) and has an effect opposite to ongoing ice accumulation that scales with water ice density (920 kg m$^{-3}$). Given the overall purity of the north polar layered deposits[5], dust accumulation is not accounted for here. Erosion rates that have been measured to be small, $2 \times 10^{-4}$ mm per year (ref. 50), are also neglected. Owing to its higher elevation, the south polar cap is expected to only accumulate small amount of dry ice[51], which is here neglected.

Because of a lack of a clear trend in our residual gravity analyses, we here assume that the gravitational potential from ongoing subsidence cannot be higher than from the polar cap accumulation rate. This indicates that ongoing subsidence rates must be at most around 26% (920/3,500) that of ice accumulation. Ice-accumulation estimates from previous work therefore limit the subsidence rate to less than 0.13 mm per year (ref. 39) or less than 0.28 mm per year (ref. 23) and we here use the minimum of these two values to limit our parameter space (Fig. 3). Choosing the other value has no effect on our derived viscosity structure, but would provide a lower limit to the polar cap age of 0.9 Ma instead of 1.7 Ma.

## Moment rate inversion from InSight

The InSight seismometer has detected dozens of marsquakes over its 4 years of activity[24], but none from the north polar regions (75–90° N). Analyses of InSight data have shown that a 3.8 to 3.9 magnitude event originating from these northern regions could have been detected[10]. Because the seismic moment rate depends on the strain rate experienced by the seismically competent lithosphere[40,41], it is possible to invert for a maximum strain rate in the northern regions based on this non-observation. The maximum strain rate is obtained by comparing the seismic moment rate ($\dot{M}_{seismic}$), as measured by InSight, to a geodetic moment rate ($\dot{M}_{geodetic}$) obtained from the Kostrov equation[40].

Mars's seismic moment rate based on the InSight catalogue can be defined in terms of the $b$ value[10] as

$$\dot{M}_{seismic} = \frac{1}{n}\frac{\Gamma(2-2b/3)}{1-2b/3}10^{9.1+3M_g/2}, \tag{5}$$

where $n$ is the number of terrestrial years, $M_g$ is the magnitude of the seismic event and $\Gamma$ is the gamma function. The geodetic moment rate, which depends on the strain rate originating from lithospheric deformation, $\dot{\varepsilon}$, scales as

$$\dot{M}_{geodetic} = 2\dot{\varepsilon}\alpha\mu V. \tag{6}$$

In this equation, $\alpha$ is a seismic efficiency factor ranging from 0 to 1, $V$ is the seismogenic volume and $\mu$ is the average shear modulus throughout this volume.

Given the non-observation of 3.9 magnitude marsquakes at high latitudes over the 4-year lifetime of the InSight mission, substituting $\dot{M}_{geodetic}$ and $\dot{M}_{seismic}$ in the above equations can be used to infer a maximum strain rate as a function of $\alpha$, $\mu$ and $V$. The shear modulus $\mu$ is obtained from the interior models and the seismogenic volume $V$ is calculated assuming a seismogenic depth given by the 573 or 1,073 K mantle isotherms[52]. In our models, $\mu$ and $V$ are found to range from 44–72 GPa and 67–334 km, with a low shear modulus corresponding to models with a thin seismogenic layer thickness. The seismic efficiency factor, $\alpha$, however, is mostly unknown for Mars, although it has been shown to be less than 0.7 (ref. 10). As a result, we consider this parameter to be 0.7 or 0.03, the latter of which is the minimum value that has been estimated on Earth[41].

We derive the maximum strain rate from the non-observation of marsquakes in the northern regions over 4 years of InSight data collection to be $1.84 \times 10^{-18}$ s$^{-1}$. As shown in Fig. 3, young polar caps (less than 1 Ma) show strain rates that are higher and inconsistent with the InSight-derived moment rate, and these models are thus excluded.

## Data availability

Interior models are available at Zenodo (https://doi.org/10.5281/zenodo.13710168)[53]. Radar data are available from ref. 6.

## Code availability

ALMA is available at https://github.com/danielemelini/ALMA3 and Perple_X is available at https://www.perplex.ethz.ch/. The GAIA code is a proprietary code of the German Aerospace Center (DLR). Users interested in working with it should contact A.-C.P. (ana.plesa@dlr.de) and C. Hüttig (christian.huettig@dlr.de).

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

**Acknowledgements** A.B. is funded by the Alexander von Humboldt Foundation. A.B., A.-C.P. and V.K. thank Geo.X—The Research Network for Geosciences in Berlin and Potsdam for providing financial support for the 'Grow Your Idea!' workshops.

**Author contributions** A.B. conceptualized the work and methodology, carried out the data analyses and modelling. A.-C.P. ran the thermal evolution simulations. V.K. and B.C.R. contributed to the viscoelastic deformation modelling. A.G. contributed to the reanalysis of the gravity signal associated with ice accumulation and isostatic adjustment rates. M.K. contributed to the seismic and geodetic moment analyses. All authors contributed to interpreting the results and writing the manuscript.

**Funding** Open access funding provided by Deutsches Zentrum für Luft- und Raumfahrt e.v. (DLR).

**Competing interests** The authors declare no competing interests.

**Additional information**

**Correspondence and requests for materials** should be addressed to A. Broquet.

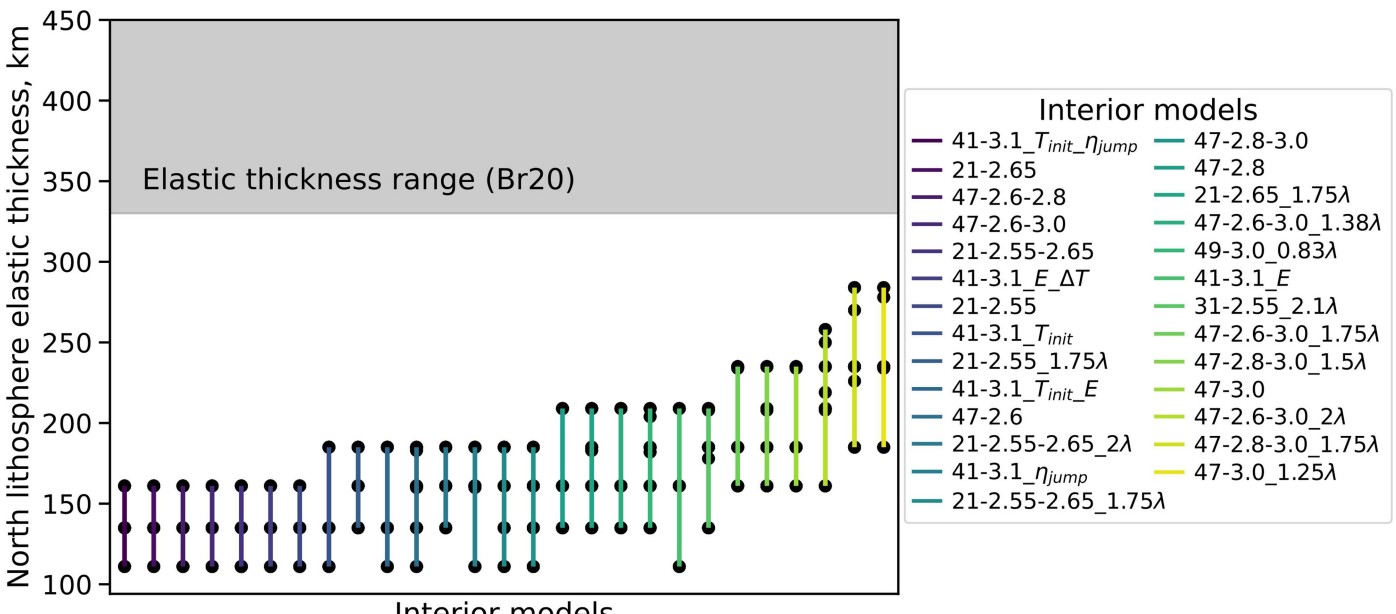

**Extended Data Fig. 1 | Elastic lithosphere thicknesses of the 27 selected interior models.** The grey shade indicates the allowed elastic thicknesses from radar analyses (Br20, ref. 6) and the black dots provide the range of possible elastic thicknesses. Line colours and interior models are the same as in Fig. 2. Interior model names are described in the Methods and Extended Data Table 1.

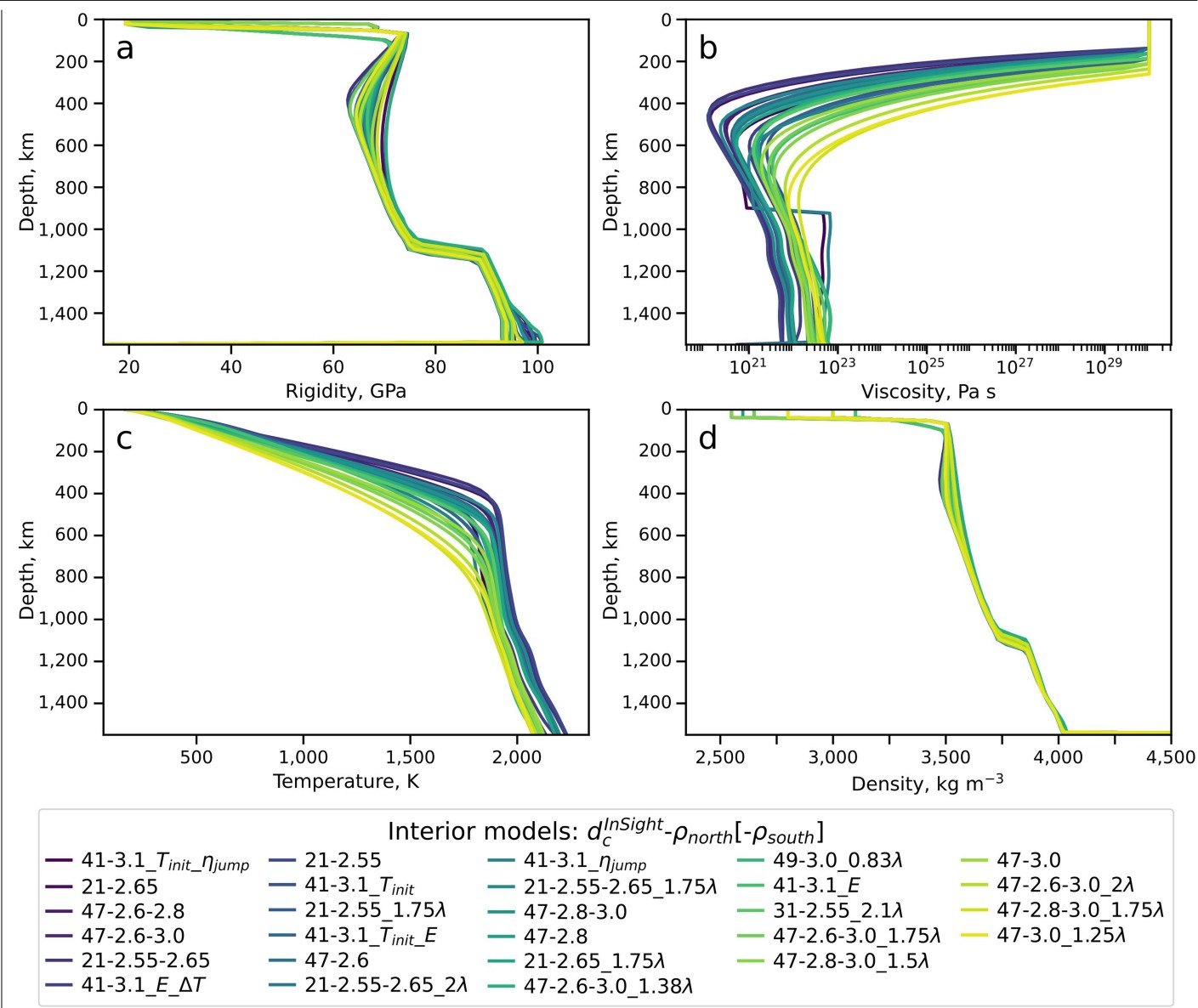

**Extended Data Fig. 2 | Present-day average interior structure of the northern regions (poleward of 60°N).** Rigidity (a), viscosity (b), temperature (c), and density (d) of the interior as a function of depth. Line colours are the same as in Fig. 2 and Extended Data Fig. 1.

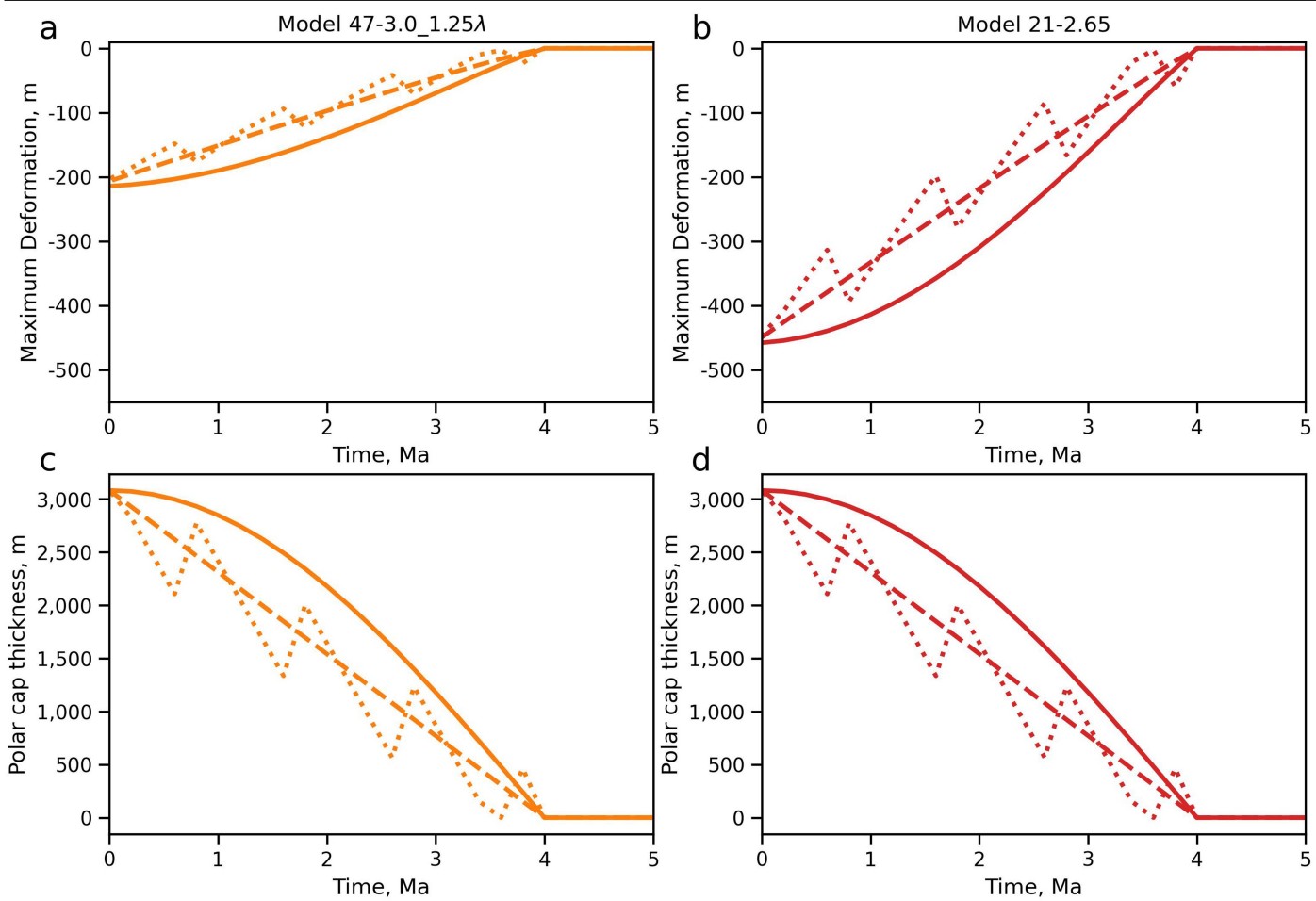

**Extended Data Fig. 3 | Effect of the ice-accumulation history on polar deformations.** Time evolution of the maximum deformation beneath the north polar cap (a, b) considering different ice-accumulation histories (linear, sinusoidal, sawtooth; c, d) for the 47-3.0_1.25λ (left) and 21-2.65 (right) models. The accumulation histories in c and d are similar.

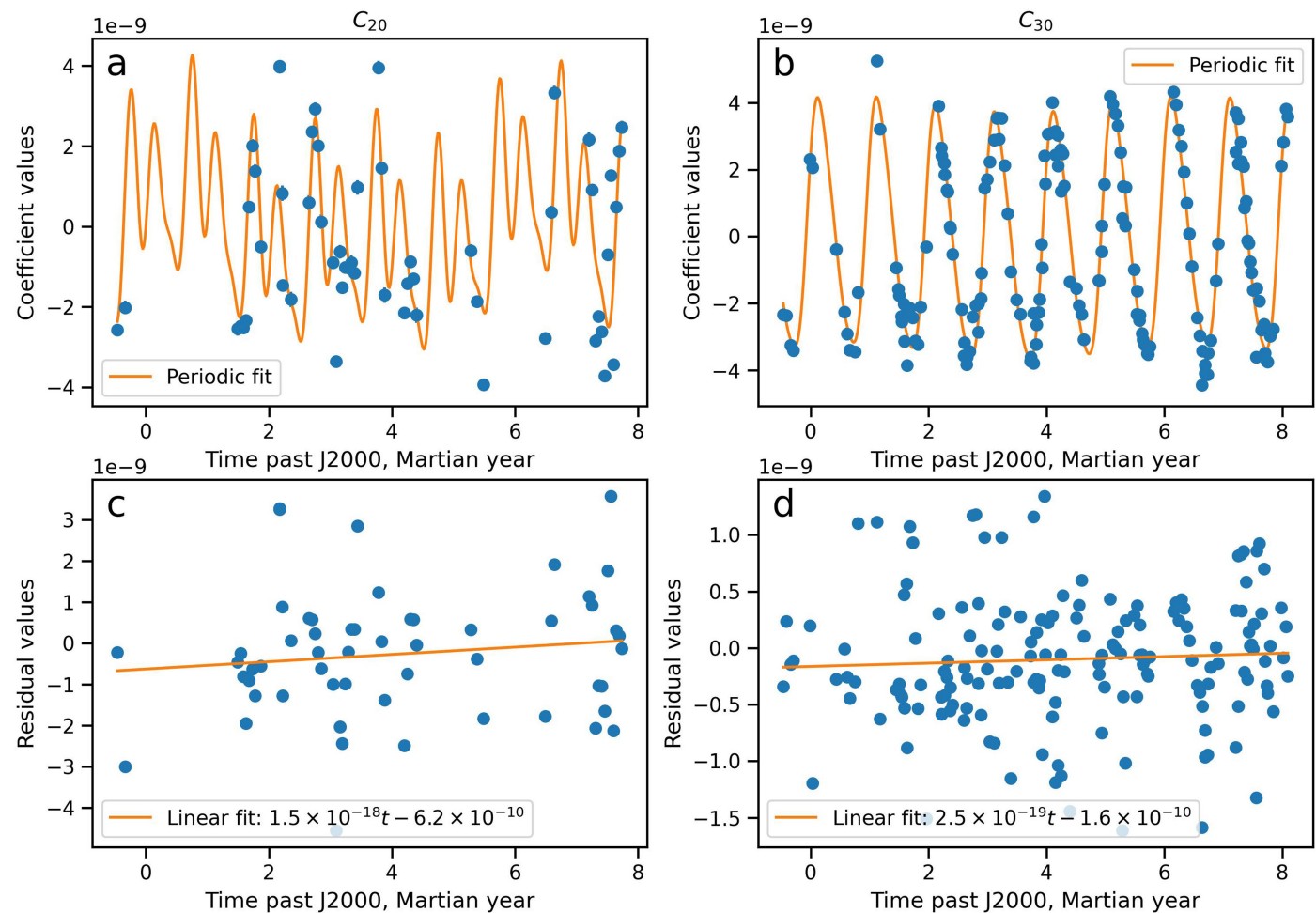

**Extended Data Fig. 4 | Time variable zonal degree 2 and 3 coefficients of the gravitational potential.** Observed values and periodic fit (a, b). Residuals from the fit and linear trend in the residuals (c, d).

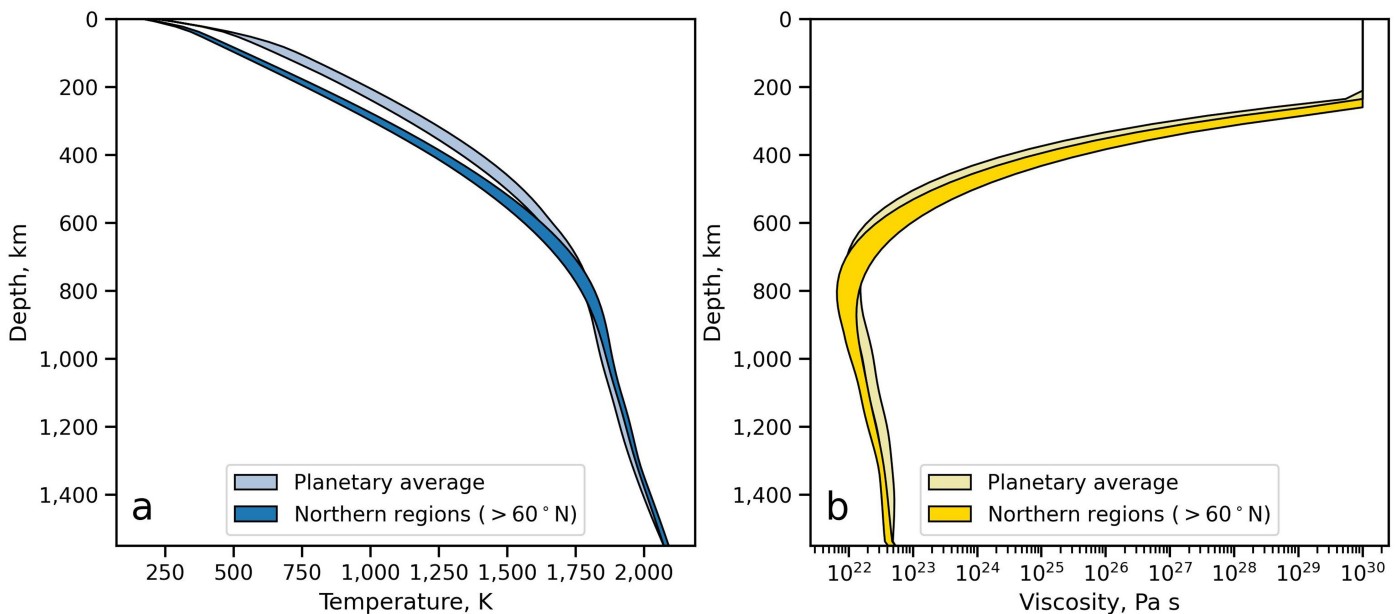

**Extended Data Fig. 5 | Constrained interior temperature and viscosity.** Interior temperature (a) and viscosity (b) ranges of the northern regions (poleward of 60°N) and planetary average for accepted models.

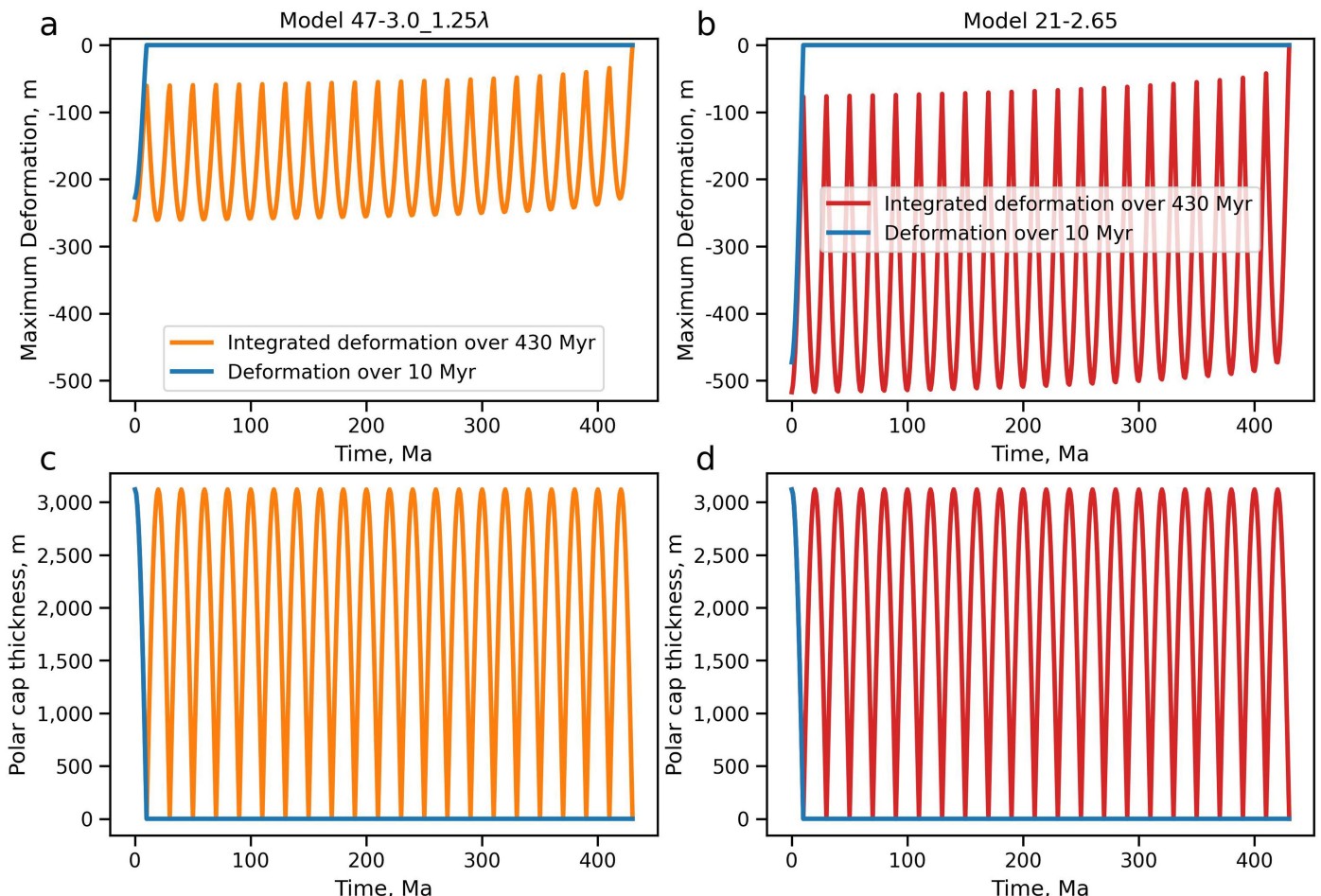

**Extended Data Fig. 6 | Effect of a periodic ice-accumulation history on polar deformations.** Time evolution of the maximum deformation beneath the north polar cap (a, b) considering a 10 Myr periodic polar cap loading (c, d) for the 47-3.0_1.25λ (left) and 21-2.65 (right) models. The accumulation histories in c and d are similar. The blue line provides a reference model with ice cap age of 10 Ma that neglects the past ice history.

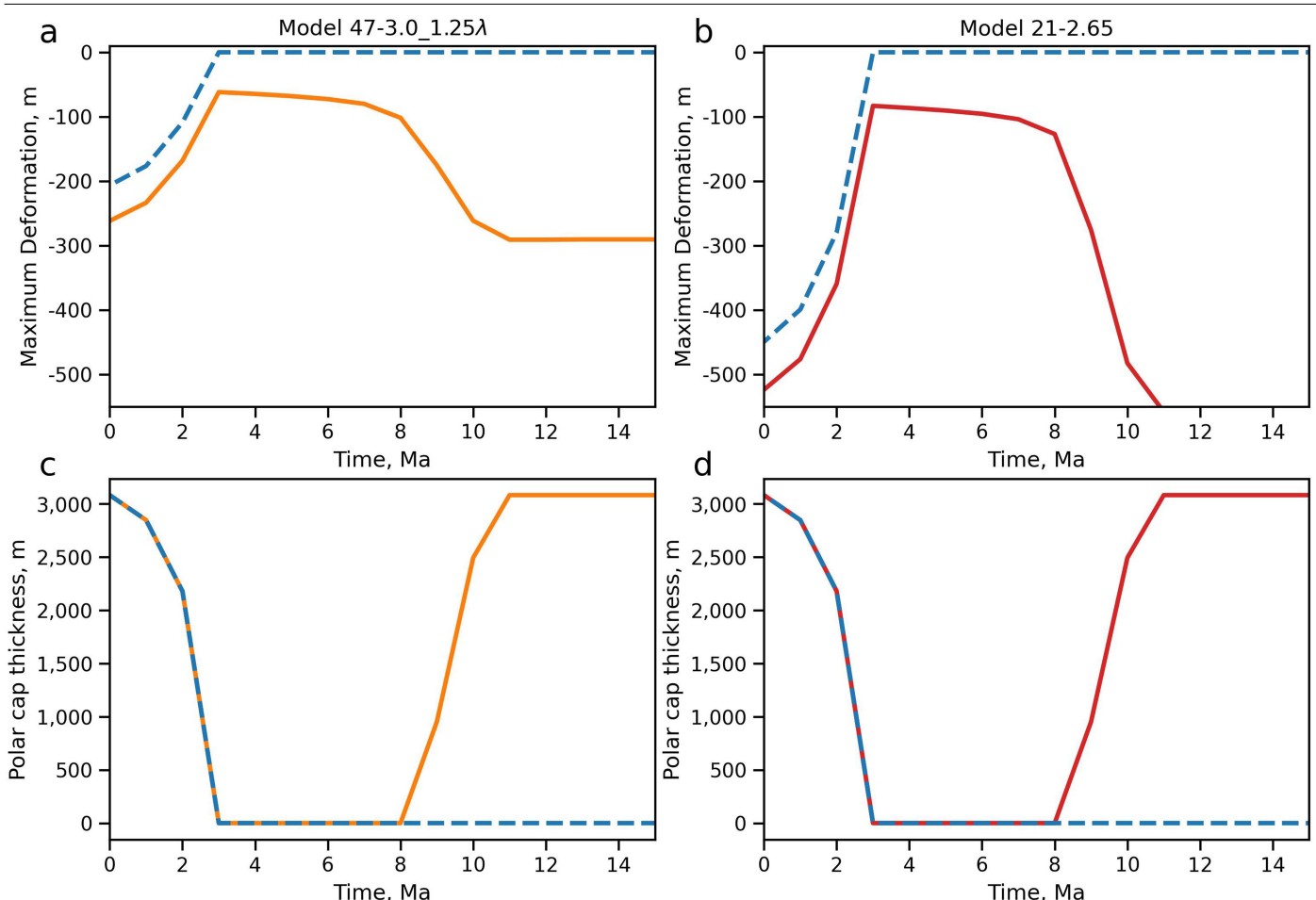

**Extended Data Fig. 7 | Effect of the ice-accumulation history on polar deformations.** Time evolution of the maximum deformation beneath the north polar cap (a, b) considering the loading history of the standard model in ref. 23 (c, d) for the 47-3.0_1.25λ (left) and 21-2.65 (right) models. The accumulation histories in c and d are similar. In this model, the past ice cap that decayed at ~8 Ma is assumed to have been long-lasting and stable over the last 500 Ma. The dashed line provides a reference model that neglects this past ice cap.

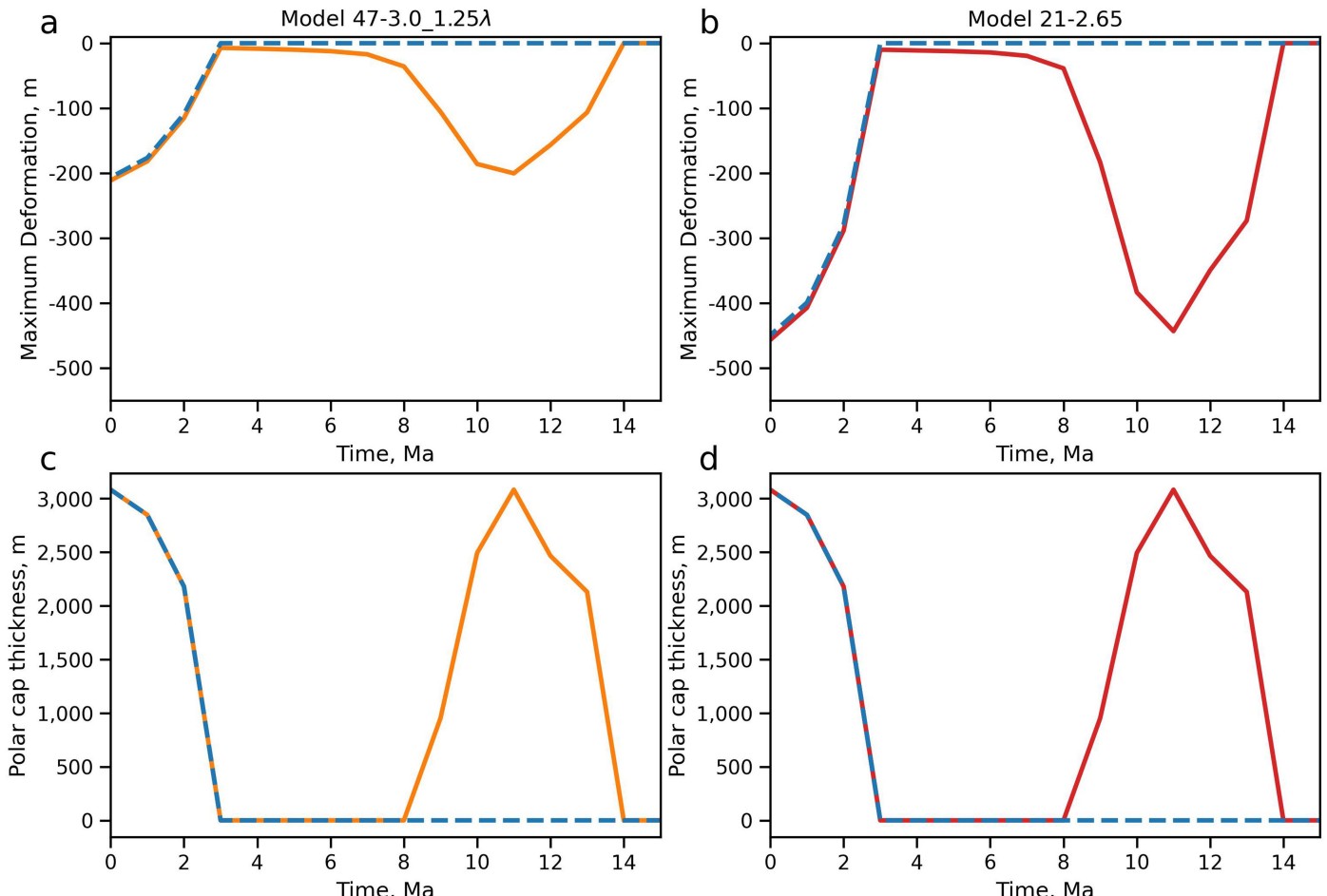

**Extended Data Fig. 8 | Effect of the ice-accumulation history on polar deformations.** Time evolution of the maximum deformation beneath the north polar cap (a, b) considering the loading history of the standard model in ref. 23 (c, d) for the 47-3.0_1.25λ (left) and 21-2.65 (right) models. The accumulation histories in c and d are similar. In this model, the past ice cap that decayed at ~8 Ma is assumed to have formed at 14 Ma. The dashed line provides a reference model that neglects this past ice cap.

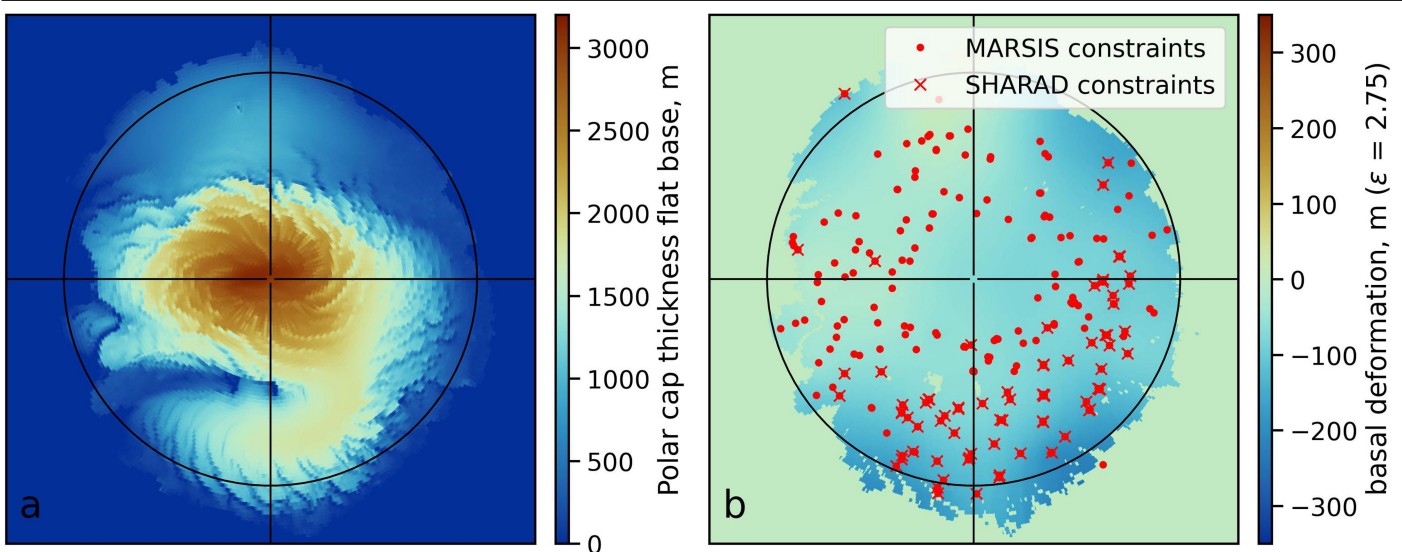

**Extended Data Fig. 9 | Polar cap thickness and basement shape.** Thickness of the polar cap assuming a flat basement (a) and deformed basement of the polar cap assuming a real dielectric constant of 2.75 in our radar measurements (b).

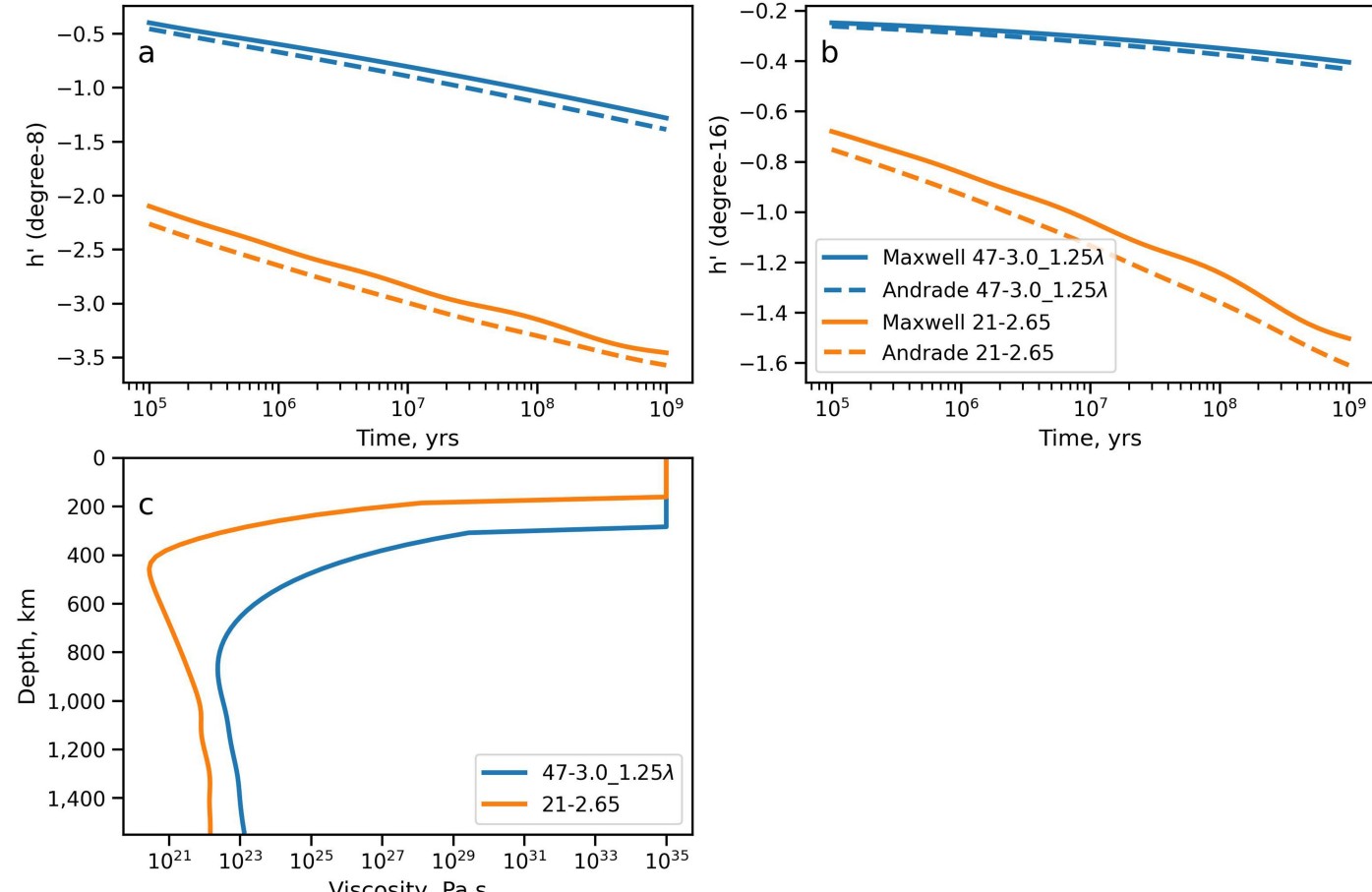

**Extended Data Fig. 10 | Love numbers for Maxwell and Andrade rheologies.** Mars interior response for Maxwell and Andrade rheology for the degree 8 (a) and 16 (b) load Love number h′ and associated interior viscosity (c) for two tested models. Love numbers in this figure are calculated assuming a Heaviside load history. The Andrade rheology is defined following ref. 8 and assumes an Andrade creep parameter of 0.3.

**Extended Data Table 1 | Summary of tested models. If not indicated, nominal parameters are initial temperature of $T_{init}$ = 1650 K, activation energy of $E$ = 300 kJ mol$^{-1}$, temperature difference across the mantle of $\Delta T$ = 2000 K**

| Interior Model | $\rho_{south}$, g cm$^{-3}$ | $\rho_{north}$, g cm$^{-3}$ | $d_c$, km | $d_c^{InSight}$, km | $\lambda$ | $\eta^{500}$, Pa s | $dT\ dz^{-1}$, K km$^{-1}$ |
|---|---|---|---|---|---|---|---|
| 21-2.55 | 2.55 | 2.55 | 30 | 21 | 1.00 | 2.0e+21 | 3.2 |
| 21-2.55_1.75λ | 2.55 | 2.55 | 30 | 21 | 1.75 | 3.3e+21 | 3.1 |
| 21-2.65 | 2.65 | 2.65 | 32 | 21 | 1.00 | 2.3e+21 | 3.2 |
| 21-2.65_1.75λ | 2.65 | 2.65 | 32 | 21 | 1.75 | 4.1e+21 | 3.1 |
| 21-2.55-2.65 | 2.55 | 2.65 | 29 | 21 | 1.00 | 2.4e+21 | 3.1 |
| 21-2.55-2.65_1.75λ | 2.55 | 2.65 | 29 | 21 | 1.75 | 2.9e+21 | 3.2 |
| 21-2.55-2.65_2λ | 2.55 | 2.65 | 29 | 21 | 2.00 | 3.8e+21 | 3.1 |
| 31-2.55_2.1λ | 2.55 | 2.55 | 41 | 31 | 2.10 | 1.5e+22 | 3.0 |
| 41-3.1_$T_{init}$ | 3.10 | 3.10 | 62 | 41 | 1.00 | 1.3e+22 | 2.5 |
| 41-3.1_E | 3.10 | 3.10 | 62 | 41 | 1.00 | 1.2e+22 | 2.4 |
| 41-3.1_$T_{init}$_E | 3.10 | 3.10 | 62 | 41 | 1.00 | 8.3e+21 | 2.5 |
| 41-3.1_$\eta_{jump}$ | 3.10 | 3.10 | 62 | 41 | 1.00 | 3.0e+22 | 2.5 |
| 41-3.1_$T_{init}$_$\eta_{jump}$ | 3.10 | 3.10 | 62 | 41 | 1.00 | 2.2e+22 | 2.6 |
| 41-3.1_E_$\Delta T$ | 3.10 | 3.10 | 62 | 41 | 1.00 | 6.0e+21 | 2.6 |
| 47-2.6 | 2.60 | 2.60 | 58 | 47 | 1.00 | 4.0e+21 | 2.9 |
| 47-2.6-2.8 | 2.60 | 2.80 | 50 | 47 | 1.00 | 3.6e+21 | 3.0 |
| 47-2.8 | 2.80 | 2.80 | 61 | 47 | 1.25 | 1.9e+22 | 2.9 |
| 47-2.8-3.0 | 2.80 | 3.00 | 52 | 47 | 1.00 | 4.1e+21 | 3.0 |
| 47-3.0 | 3.00 | 3.00 | 69 | 47 | 1.00 | 8.0e+21 | 2.7 |
| 47-2.6-3.0 | 2.60 | 3.00 | 43 | 47 | 1.00 | 3.3e+21 | 3.0 |
| 47-2.6-3.0_1.38λ | 2.60 | 3.00 | 43 | 47 | 1.38 | 1.8e+22 | 2.9 |
| 47-2.6-3.0_1.75λ | 2.60 | 3.00 | 43 | 47 | 1.75 | 1.0e+22 | 2.6 |
| **47-2.6-3.0_2λ** | **2.60** | **3.00** | **43** | **47** | **2.00** | **2.5e+22** | **2.4** |
| 47-2.8-3.0_1.5λ | 2.80 | 3.00 | 52 | 47 | 1.50 | 1.2e+22 | 2.5 |
| **47-2.8-3.0_1.75λ** | **2.80** | **3.00** | **52** | **47** | **1.75** | **6.4e+22** | **2.3** |
| **47-3.0_1.25λ** | **3.00** | **3.00** | **69** | **47** | **1.25** | **4.9e+22** | **2.3** |
| 49-3.0_0.83λ | 3.00 | 3.00 | 71 | 49 | 0.83 | 1.8e+22 | 2.9 |

If noted in the interior model's name, $E$ = 325 kJ mol$^{-1}$, $T_{init}$ = 1850 K, and $\Delta T$ = 2205 K. The two models annotated with $\eta_{jump}$ have a 25-fold viscosity jump at around 900 km as suggested on Earth[54]. The heat production factor ($\lambda$) scales the crustal heat production with respect to the nominal average Gamma ray estimate of 49 pW kg$^{-1}$ (ref. 28). $d_c$ is the average crustal thickness, $d_c^{InSight}$ is the crustal thickness at the InSight landing site, $\eta^{500}$ is the volume average present-day viscosity from 500 km to the core, and $dT\ dz^{-1}$ gives the upper mantle (100–400 km) thermal gradient. The three models that fit our constraints on glacial isostatic adjustment are displayed in bold.

**Extended Data Table 2 | Summary of time-variable gravity analyses for the $C_{20}$ and $C_{30}$ coefficients**

| Coefficient | m (×10⁻¹⁸) | b (×10⁻¹⁰) | p-value | r-value |
|---|---|---|---|---|
| $C_{20}$ | 1.5±1.6 | 6.2 | 0.35 | 0.13 |
| $C_{30}$ | 0.2±0.3 | 1.6 | 0.46 | 0.06 |

Values for *m* and *b* are for the *mx+b* linear fit to the residuals. The p-value tests the null hypothesis that the residuals have no correlation with time based on Wald's test, and the r-value provides the Pearson correlation coefficient of the residuals with time.