## [Peer Review File · Nature]

Glacial isostatic adjustment reveals Mars' interior viscosity structure

Corresponding Author: Dr Adrien Broquet

**Parts of this Peer Review File have been redacted as indicated to remove third-party material.
This file contains all reviewer reports in order by version, followed by all author rebuttals in order by version.**

Version 0:

Reviewer comments:

Referee #1

(Remarks to the Author)
Review of manuscript Nature # 2024-08-17425

Dear Editor,

This work explores the viscosity structure of Mars using evidence from the post glacial deformation induced by the north polar cap and a number of geophysical constraints. The main result is that the viscosity of Mars is relatively high (compared to Earth), exceeding 10^{22} Pa.s (or 10^{23}) and the crust is relatively stiff on average (> 40 km).

The findings of this work are important because they sheds new light on the internal dynamics of the planet. As far as I know it is the first time that these fundamental parameters of Mars' structure are obtained using Glacial Isostatic Adjustment modeling and evidence from the polar regions.

It is an ambitious work, particularly well organized and well presented, based upon a quite robust set of geophysical constraints and appropriate methods, which merits publication after some points have been addressed and better clarified.

As far as I can see, the statistics and the uncertainties are dealt with correctly and the conclusions of this work are motivated by the existing evidence.

I have a few points section-by-section that may be useful to improve the manuscript.

- Abstract

1- Very informative and to-the-point, appropriate. The fact that 'the negligible flexure' beneath the ice cap seen by radars can help to constrain viscosity, may sound somewhat strange. Negligible flexure could indicate a rigid planet (or an extremely high viscous planet if you want, which is after all consistent with the main finding here). This point should be better clarified in the pertinent sections.

- Introduction

2- This section is well written and the motivations clearly stated by enlightening the limitations of previous work. The fact that in this work one-dimensional models for the martian interior are adopted should be explicitly stated since it is, by itself, a significant limitation.

3- I am also a bit surprised to see that the authors intend to used an Andrade rheology in view of the long time scales of evolution of the polar cap (see also my comment to Figure S3 below). Andrade can be left here and throughout the paper, but its introduction seems a bit specious.

- Mar's interior structure from geodynamic models

4- Are the models tested by using program GAIA viscoelastic or simply viscous?

5- All the 27 models retained after GAIA analysis are characterized by an inflection in the viscosity profiles, a sort of broad low-viscosity layer, in the range between 1000 and 400 km depth (left frame of Figure 2). What is ultimately the origin of this feature? At least to me this is not clear from the arguments given.

6- I am surprised to see that GAIA provides absolute viscosity values and not relative values. Has viscosity of one of the layers been fixed arbitrarily a-priori?

- Constraints on viscoelastic deformation of the northern region

7- A long Section that could be better organized in sub-sections, one for each constraint, in order to facilitate the readers.

8- The sentence "We note the changing the ice accumulation history ... has negligible effects ... viscosity structure" would require some clarification. It is also unclear, at this stage, what is the uncertainty on the mantle viscosity structure could be tolerated in this study. The uncertainty on the range of viscosity quoted in the abstract is very high, being 10^{22} Pa.s (or may be 23) a lower bound, apparently.

9- Incompressibility in ALMA. I fully agree with what the authors say about compressibility. However, are the GAIA computations compressible or not? If they are, there is a slight consistency problem.

10- The estimated downward deformation in the northern regions <0.13 mm/year is surprisingly precise, in view of all the sources of uncertainty, even after reading the pertinent supplementary material (methods). I am also surprised to see (methods) that other possible subsidence rates (the authors quote <0.28 mm/year) would not affect the derived viscosity structure. Why?

11- The argument of non-observation of "marsquakes" as a support to low strains in the northern region is somewhat debatable. I would say that the InSight evidence is not in contrast with the gravity analysis, not that 'it remains consistent' with it.

- Discussion

12- I like this section but would recommend a Conclusions section, too. To improve the presentation, the material presented here could be organized in two sections... or may be the whole section could be entitled "Discussion and Conclusions", with the latter reported in the very last paragraph.

- Minor points, with suggestions for improvements:

13- Figures. The figures quality is good but it is not extraordinary.

14- Figure 2. Here an in other instances one wonders what is the meaning of the labels adopted to indicate the models used. Ah, I see that this is given in the caption of Table S1. Well, why not labeling these models as (for example) M1, M2, M3, ... in Table 1 so that in Figure 2 you can just reference M1, M2, M3... It is just a suggestion, of course.

15- Table S1. Looking at the actual values of the viscosities for the three starred models (fitting models), I would say more $>=10^{23}$ Pa.s than $>10^{22}$ Pa.s as reported in the abstract and in the Discussion section.

16- Figure S1. Ditto. See above my Figure 2 comment on the models names.

17- Figure S2. What is the meaning of the colors employed to draw these curves? Please add a color table or some information in the caption of the figure.

18- Figure S3. What is the ice time history adopted for the plot of the Love numbers? This is not specified and unclear. Does "Andrade" only include an elastic and a transient power law in the viscoelastic creep function, or also a Newtonian element?

19- Figure S5. In a) and b), please use the same y-range so to allow a fair comparison between the plots.

20- Figures S6 and S7. The same as in Figure S5 above.

21- References. There are capitalization problems in a significant number of papers titles, please check. Beside this, I think that the paper is giving appropriate credit to previous work.

I hope these comments shall be useful to improve this very good contribution.

Referee #2

(Remarks to the Author)

Review of Glacial isostatic adjustment reveals Mars' interior viscosity structure

This paper updates existing models of crustal deformation and viscosity using new input from the time-variable gravity field

and the seismic moment rate determined from InSight. As past groups have done, much of this analysis relies on results from MARSIS and SHARAD plus the MOLA instrument. The addition of these new constraints allows the authors to better constrain the geophysical properties of the crust and mantle of Mars along with estimate a range of ages for the north polar ice cap.

From the north polar ice perspective, I find the results broadly consistent with established literature, and I believe that they will increase the fidelity of arguments that the north polar ice is relatively young, only a few to 10 million years old. This work may appear incremental; however, the inclusion of new datasets and constraints is quite novel. I believe this is a valuable contribution.

My expertise are not in line with the modeling work here, so I focus my review to help strengthen the arguments and support consistency, especially with citations, which are sometimes misleading, incorrect, or out of date. I have no major comments, but there two moderate comments and several opportunities for minor modifications that can strengthen the manuscript.

Two moderate comments are that 1) the authors provide ranges but not best fits. Obviously, this type of analysis inherently comes with error bars, and it is important to show them or provide valid ranges that fit the given data. That said, it would be useful to the community of researchers who will read this paper to know the best fit, given the new inputs to the model.

And 2) The manuscript does not compare the new results to past results. It kind of pretends that past work of this type does not exist, and so readers are left wondering how different is this than the Phillips 2008 results.

-Isaac Smith

Unfortunately, there are no line numbers, so I have to quote directly from the text

Paragraph 3, "ongoing plume activity (Fig S1)."

-Figure S1 does not mention plume activity, so some change should be made here

"suggest that the current northern ice cap formed over the past 4 Ma, with ice-sheet thickness increasing linearly in response to a gradual decline in polar insolation²³".

—Reference 23 does not provide a linear thickness increase. It is more sawtooth. Greve et al 2006 modeled the same sawtooth increase in thickness, and Smith et al., 2016 demonstrated that the stratigraphy includes unconformities that represent periods of material loss between accumulation. Please rephrase. This issue shows up again in the following paragraph

"Because the age and long-term loading history of the northern ice cap are poorly known, we assume a linear increase in ice-sheet thickness over time, leading to its current state"

"This effect would prominently affect all estimates of the strength of the elastic lithosphere and planetary thermal state that have neglected viscous relaxation^{5, 6}."

-this is misleading. In the introduction for Phillips 2008, they said "Alternatively, the response to the load may be in a transient state controlled by mantle viscosity"

"surface age (100 kyr, ref35)"

-this is an out of date surface age. A much more recent paper suggests 1.5 kyr for the surface age: Landis 2016

"We note that changing the ice accumulation history from linear to exponential or sinusoidal in our viscoelastic calculations has a negligible effect compared to the uncertainty in the mantle viscosity structure."

-I think you are probably right, but this is not established in the manuscript. Perhaps you could add more in supplemental information. Readers will wish to see how these changes affect the outcome, especially give than ice accumulation has not been linear.

"Changes in the seasonal cap whose load is 3 orders of magnitude lower than the perennial cap"

-The authors are ignoring past episodes of atmospheric collapse that could have added much more CO2 mass load. Please see Buhler et al., 2019, 2021 or Bierson 2016 and Manning 2016

"The present-day north polar cap has a thickness of about 3.1 km and bulk density of about $1200 \pm 300 \text{ kg m}^{-3}$, as constrained from radar analyses^{6, 34}"

-34 doesn't mention density. Is this a correct citation?

"Because of its higher elevation, the south polar cap is expected to only accumulate small amount of dry ice³⁹, which are here neglected"

-there are newer and better references for this statement, e.g. Thomas, P.C., Calvin, W., Cantor, B., Haberle, R., James, P.B., Lee, S.W., 2016. Mass balance of Mars' residual south polar cap from CTX images and other data. *Icarus* 268, 118–130. <https://doi.org/10.1016/j.icarus.2015.12.038>

"If the northern strain rates were too large, crustal failure and faulting would occur, potentially leading to the detection of a marsquake by InSight"

-maybe this is the better argument - that faults are not observed. Can the authors say anything about the size of expected faults if the crust were to fail brittlely?

"Based on predicted magnitudes from the InSight catalogue, a 3.8 to 3.9 magnitude event originating from the polar regions (75–90°N) could be detectable¹⁰. Thus, the non-observation of such marsquakes during the InSight mission can be used"

-tricky argument given that InSight was not active for long. Can the authors say statically how many magnitude 3.9 earthquakes would be expected during the lifetime of InSight? Without that, this argument is meaningless. Say, we only expect one every 5 years in the north polar region, and InSight missed it, would that tell us anything?

"One way to help reduce the ongoing deformation, and thus our inferred interior viscosity, would require the present-day ice cap to have formed on the sedimentary-infilled flexural trough of a former ice cap. In that framework, the ongoing downward deformation would compete with the rebound related to the past ice-sheet. However, while such sedimentary infilling may be related to the basal unit beneath the central portion of the polar layered deposits³⁴, it is not seen beneath the Gemina Lingula region near 80°N (Fig. 1). Therefore, this competing effect alone cannot account for the negligible flexure observed in that region."

-I like that you brought this up. Would it be possible to assume that the basal unit (which Nerozzi found the volume for) is 1 Ga old (Tanaka 2005?) and rerun these models? I think the distribution of basal unit is publicly available. I would be curious about how much this lowers the upper age limit. Please state that here.

"MARSIS radar analyses were performed at 213 locations spatially scattered across the polar deposits. All available MARSIS radargrams close to each location were investigated, the reflections arising from the icy surface and the ice/substratum interface were visually identified. Using the same framework, analyses of SHARAD radargrams were performed at 78 locations."

-I don't think Figure 1b shows 300 locations. Please clarify in the text or figure caption.

"The radar thickness depends on the dielectric properties of the icy materials, which are unknown."

-This is a rather misleading statement. Grima et al., 2009 completed an excellent analysis that the authors are ignoring here. That paper is a critical data point that has been cited hundreds of times. Ignoring it here does allow the authors to invent a number to use for ϵ' , but that number is not realistic. I understand that their analysis does allow values from 2.5 to 3.5, and that range may have value for their model, but as an avid researcher of north polar ice, I am very interested to hear what choosing a value of 3.1 or 3.15 would do to the results.

Same with the following a little lower in the text

"For context Fig. 1b shows a flexure map beneath the northern cap that was obtained by assuming a dielectric constant of 3.0 for our MARSIS and SHARAD radar measurements."

In the section about past ice caps

"This model starts at 10 Ma with a past northern ice sheet that disappears at about 8 Ma. After a time-interval with no northern ice cap, a new northern ice sheet forms with a linear increase in thickness from 4 Ma to present-day (Fig. S6). A key outstanding unknown is what was the state of the 10 Ma ice sheet and whether it was long-standing and at equilibrium or more recently formed. In one model, we assume that the previous ice cap was present over the last 500 Ma and thus at equilibrium (Fig. S6) and in a second model, we assume that it formed at 14 Ma (Fig. S7). The state of this past ice sheet drastically affects its influence on the present-day deformation. In the case where the ice cap was long-lasting, the present-day deformation is increased by up to 70 m, whereas"

-Why does the analysis ignore the basal unit that has a significant volume and mass over 10^9 years?

Figure 1

Panel B: how many points are included?

How was panel c made? There is no attribution, although I expect it was pulled from a past publication. Also, 3.0 is inconsistent with Grima 2009.

And, to support the argument here about deflection, there may be value in extending the radar image further beyond the layered ice (south in both directions) to demonstrate that the lower boundary is coplanar to the surrounding terrain

Figure S4

The color scale here is hard to see, please choose different colors. I have no difficulty distinguishing colors, but people with vision impairment will not be able to tell the difference between blue with black border and blue with purple border. Same with yellow. This really needs to be changed to distinguish between the planetary average and the polar region. Panel b does not have a color legend.

Please check all of the other figures for color legends and enough annotation to interpret them.

Version 1:

Reviewer comments:

Referee #1

(Remarks to the Author)

Dear Editor

I have carefully considered the authors' response to my queries. I think the paper is acceptable now.

Referee #2

(Remarks to the Author)

The authors have satisfactorily addressed my comments.

Isaac Smith

We thank the editor for the time taken in assessing our manuscript and highly appreciate the reviews from both reviewers. While all of the reviewers' comments are addressed point by point further below, we here quickly summarize the main aspect of each review and how our materials were modified.

Reviewer #1 is supportive of the study and its main conclusions and highlights the novelty of the work. No major comments were raised, but several suggestions were made to improve the clarity of the text and quality of the Figures. All have been considered in the revised manuscript.

Reviewer #2 is also supportive of the work, but 1) suggests that best-fits be presented in addition to the currently presented ranges and 2) has some reservations considering how prior work, in particular Phillips et al. (2008) [cited in the manuscript], are referenced. Regarding point 1), our study currently provides a range of equally plausible solutions. Multiple datasets and techniques have been used, each carrying their own uncertainties and assumptions. For that reason, the uncertainty in accepted models cannot be simply represented by a Gaussian. This implies that a single statistically significant best-fit model cannot be defined. Nevertheless, we have added below some information on what parameters would be obtained considering the radar best-fit only, but note that these should not be particularly preferred. Regarding point 2), we strongly emphasize that our work drastically differs from that presented in Phillips, which did not compute viscoelastic deformation and did not jointly estimate the age of the polar cap and interior structure of Mars. We have clarified some statements to better highlight that point. Finally, we have designed a new model considering a 1 Ga basal unit when assessing the present-day deformation (see Figure pasted below), and have also added a supplementary Figure (S3) to show the influence of the ice accumulation history. Both have minor effects on the deformation when compared to the interior structure.

We have pasted below in black all comments from the editor and reviewers. Our replies are shown in blue.

Editor

Dear Adrien,

Your manuscript entitled "Glacial isostatic adjustment reveals Mars' interior viscosity structure" has now been seen by two referees, whose reports are attached below. Please allow me to apologise for the delay in obtaining these comments. While both referees find your work of interest, they have raised points that need to be addressed, in the form of a revised manuscript, before we can make a final decision on publication (which we make in consultation with the referees).

In addition to addressing the referees' comments, there are several editorial points that we will also need for you to address when revising your manuscript -- although your paper is generally

in good shape.

Length: The present text length of your manuscript is somewhat longer than our usual length limits for a short *Nature* Article. We do have some flexibility and could allow you to maintain the present main text length, however we will need to consider this a firm upper limit. Please also include a copy of your text in MS Word format, if possible, and include line numbering when submitting your revised manuscript.

We have removed several sentences and words to reduce the text length. In the revised version of the main text, which addresses the reviewers' comments, the word count was decreased from 2735 to 2646.

We have also converted our manuscript to MS Word format and included line numbers.

Sub-headings: In our new all-Article format, we do encourage several sub-headings within the main text of short Articles to break up the text. Such sub-headings, however, may only be up to 40 characters in length, including spaces. Also, please avoid generic sub-headings such as 'Discussion' or 'Conclusions', and instead indicate the specific content of each section.

We have changed the "Discussion" sub-heading to "Highly viscous mantle and young ice cap", which better indicates the specific content of the section. Based on the reviewers' comments, we have also added several sub-headings to improve clarity. All are ≤ 40 characters in length.

Methods: At the end of the main text document (after the main figure legends), there should be a section entitled "Methods", which provides a more detailed discussion of the additional methodological information that would allow other researchers to replicate the results (we define "Methods" quite broadly, so this is not limited to details of experimental protocols – supplementary discussion and analysis can also be included). The Methods section will not appear in the print version but will be fully copy-edited and appear online in the full-text HTML and PDF versions. The Methods section should be written as concisely as possible but should contain all elements necessary to allow interpretation and reproduction of the results. If there are additional references in the Methods section, their numbering should continue from the last reference in the main paper, and the list should follow the Methods section.

We have moved the Methods section to after the main Figures' caption.

Main Text Statements: We require authors to provide a detailed Author Contribution statement immediately after the acknowledgements; the specific contributions of each author must be listed. It is also a condition of publication that authors include an Author Information statement indicating how to access information regarding reprints and permissions, stating whether or not there is a financial or non-financial competing interest, and naming the author to whom correspondence and requests for materials should be addressed. Please ensure that this section is included in the manuscript file after the Methods (but before the Extended Data legends) - it will not appear in the print version but will appear online in the full-text HTML and PDF

versions. For details of "end note" style and an example see <https://www.nature.com/nature/for-authors/formatting-guide>.

We have added statements regarding: Supplementary information, Correspondence, and Reprints and Permission as required.

Display items: We ask that you take stock of all the data that have been generated throughout the review process and ensure that only the data most central to the conclusions are presented in the main figures. Figures should be comprehensible to readers in other or related disciplines, and assist their understanding of the paper. We encourage authors who are describing complex processes to include a schematic of the main finding as part of the Extended Data to aid readers unfamiliar with the immediate discipline. Figures should be as small and simple as is compatible with clarity. All panels of a figure should be logically connected; each panel of a multipart figure should be sized so that the whole figure can be reduced by the same amount and reproduced on the printed page at the smallest size at which essential details are visible. For guidance, *Nature's* standard figure sizes are 89 mm (one column), 120 mm (one and a half columns), or exceptionally 183 mm (two columns) wide; the full depth of a *Nature* page is 247 mm. All panels of figures should be presented on a single page and assembled into a rectangular shape for publication; please indicate any essential alignments (parts horizontal, vertical, spacings of stereo pairs, etc.). Tables should be prepared using the Table menu in Microsoft Word.

Figure formatting: Lettering in all figures (labelling of axes and so on) should be in uniform, sans-serif font, in lower-case type, and large enough to permit substantial reduction for publication (minimum font size 5 pt). Separate parts of a figure are labelled a, b, etc. Units have a single space between the number and the unit, and follow SI nomenclature or the nomenclature common to a particular field. Thousands are separated by commas (1,000). Unusual units or abbreviations are defined in the legend. Scale bars rather than magnification factors should be used.

We have changed main text figures such that thousands are comma-separated, as required.

Referee #1:

This work explores the viscosity structure of Mars using evidence from the post glacial deformation induced by the north polar cap and a number of geophysical constraints. The main result is that the viscosity of Mars is relatively high (compared to Earth), exceeding 10^{22} Pa.s (or 10^{23}) and the crust is relatively stiff on average (> 40 km).

The findings of this work are important because they sheds new light on the internal dynamics of the planet. As far as I know it is the first time that these fundamental parameters of Mars' structure are obtained using Glacial Isostatic Adjustment modeling and evidence from the polar

regions.

It is an ambitious work, particularly well organized and well presented, based upon a quite robust set of geophysical constraints and appropriate methods, which merits publication after some points have been addressed and better clarified.

As far as I can see, the statistics and the uncertainties are dealt with correctly and the conclusions of this work are motivated by the existing evidence.

I have a few points section-by-section that may be useful to improve the manuscript.

We greatly appreciate the positive feedback on our work. All comments definitively helped us improve the manuscript and are addressed in detail below.

- Abstract

1- Very informative and to-the-point, appropriate. The fact that 'the negligible flexure' beneath the ice cap seen by radars can help to constrain viscosity, may sound somewhat strange. Negligible flexure could indicate a rigid planet (or an extremely high viscous planet if you want, which is after all consistent with the main finding here). This point should be better clarified in the pertinent sections.

Thank you. We would like to emphasize that we do more quantitative radar analysis than suggested in our summarizing abstract statement "negligible flexure". From radar data, we have some constraints with error bars on the amount of deformation. Within error bar flexure could be near-zero, but it could also be a few hundreds of meters. This means that within uncertainties, the interior mantle can be fully elastic or highly viscous.

- Introduction

2- This section is well written and the motivations clearly stated by enlightening the limitations of previous work. The fact that in this work one-dimensional models for the martian interior are adopted should be explicitly stated since it is, by itself, a significant limitation.

The reviewer raises a good point here. However, it is important to note that our thermal evolution models are 3-D with spatial variations in mantle temperature and hence viscosity, as well as crustal thickness that is derived from gravity/topography analyses. We realized that latter point was probably not clear and have added a sentence in the methods section (line 240):

"Our geodynamic simulations use the 3-D structure of the crust, as derived from gravity, topography and InSight data²⁶"

In addition, our models predict the interior structure of Mars (rigidity, viscosity, temperature) to be homogenous in the northern hemisphere. Given that we are interested in a local loading

event in the northern hemisphere, we do not expect large differences between our model (using 1-D profiles) compared to a more complex 3-D viscoelastic loading model. We expect these differences to be largely below the uncertainty in the interior structure as shown in Figure 2 and Figure S2. In addition, the 1-D modelling approach is computationally cheap and allowed us to explore a large number of loading scenarios, with varying interior structure and ice history. Finally, the north polar cap itself is a good candidate for 1-D models as its shape and associated load is mostly axisymmetric.

This is now noted in the Methods section (line 306):

“This study employs average 1-D interior profiles to constrain viscoelastic deformations beneath the north polar cap. While future work may achieve larger accuracy by modelling 3-D deformations from a 3-D interior structure and polar cap load, several mitigating factors should be noted. Our 3-D geodynamic models indicate that Mars’ northern regions exhibit homogeneous properties (rigidity, viscosity, temperature) comparable to the global average (Fig. S5). This indicates that lateral variations in the interior structure have minor effects on the estimated deformations. In particular, we expect these effects to be drastically lower than the current uncertainty in Mars’ interior structure (Fig. 2). Furthermore, the axisymmetric shape of the north polar cap load reduces the influence of the polar cap geometry on deformation estimates. For these reasons, differences between a full 3-D viscoelastic loading model and our current model are expected to be small.”

3- I am also a bit surprised to see that the authors intend to use an Andrade rheology in view of the long time scales of evolution of the polar cap (see also my comment to Figure S3 below). Andrade can be left here and throughout the paper, but its introduction seems a bit specious.

Thank you for the comment. We initially added the Andrade rheology sensitivity analysis to demonstrate that this more complex rheology doesn’t influence our results. Andrade has gained attention in the planetary science community due to its applicability to short-term processes on planetary bodies, and we often had questions regarding the effect of choosing an Andrade rheology. Although the inference that *Andrade has no effect on long-term processes* may be straightforward for those familiar with viscoelastic modelling, we believed it would be valuable to share more information on this with the broader Mars research community.

- Mars’ interior structure from geodynamic models

4- Are the models tested by using program GAIA viscoelastic or simply viscous?

GAIA models are indeed viscous, which is now noted line 59: *“We simulate Mars’ 3-D thermal evolution and interior viscous flow using the geodynamic code GAIA”.*

5- All the 27 models retained after GAIA analysis are characterized by an inflection in the viscosity profiles, a sort of broad low-viscosity layer, in the range between 1000 and 400 km

depth (left frame of Figure 2). What is ultimately the origin of this feature? At least to me this is not clear from the arguments given.

Good question. This viscosity profile is typical for the interior of planets and the inflection occurs where the temperature profile changes the slope due to the transition from conductive to convective regions. The viscosity decreases with temperature first due to the fact that the temperature increases very fast in the top part of the planet, but in the deeper mantle, the viscosity can increase again due to the pressure dependence.

6- I am surprised to see that GAIA provides absolute viscosity values and not relative values. Has viscosity of one of the layers been fixed arbitrarily a-priori?

The reviewer is correct here. The absolute interior viscosity is defined based on a viscosity law that scales the reference viscosity (with tested values of 10^{20} – 10^{21} Pa s) and associated reference pressure and temperature. These reference values are based on experimental work and are typically used in the Mars interior community (see Plesa et al., 2022 ref 7 for more information).

We have added the following statement to clarify this (line 242):

“Absolute viscosities can be obtained using the Arrhenius viscosity law and considering reference values for the viscosity, pressure, and temperature⁷.”

- Constraints on viscoelastic deformation of the northern region

7- A long Section that could be better organized in sub-sections, one for each constraint, in order to facilitate the readers.

This is a great suggestion, thank you. The revised manuscript now includes new sub-headings:

- Ice history and viscoelastic relaxation
- Constraints from radar observations
- Mars' time variable gravity field
- Low polar strain rates
- Highly viscous mantle and young ice cap

The 4 first sub-headings replace our initial “Constraints on viscoelastic deformation of the northern region” and the last one replaces “Discussion”.

8- The sentence “We note the changing the ice accumulation history ... has negligible effects ... viscosity structure” would require some clarification. It is also unclear, at this stage, what is the uncertainty on the mantle viscosity structure could be tolerated in this study. The uncertainty on the range of viscosity quoted in the abstract is very high, being 10^{22} Pa.s (or may be 23) a lower bound, apparently.

Good point, this is an important statement that deserved more information [also noted by

reviewer 2]. We have added a supplementary Figure (now Figure S3) that shows the difference in the viscoelastic deformation for different ice accumulation histories (linear, sinusoidal, and sawtooth) considering two possible mantle viscosity profiles. This Figure shows that the ice accumulation history does affect the time-variable deformation, but does not significantly affect the recent / final deformation, which is used in our analyses.

We consider the uncertainty in the viscosity structure to be given by Figure 2 (coloured curves). In Figure S3, both models are equally possible according to earlier work. The high-viscosity model (left plots) predicts deformations of ~ 200 m, whereas the low-viscosity model (right plots) shows deformations of ~ 450 m. Thus, the effect of the ice-accumulation history on the present-day deformation (< 10 m) is drastically less important than from the uncertainty in the viscosity structure (~ 200 m).

9- Incompressibility in ALMA. I fully agree with what the authors say about compressibility. However, are the GAIA computations compressible or not? If they are, there is a slight consistency problem.

The GAIA simulations are not fully compressible, but use the Extended Boussinesq Approximation that considers some compressibility effects such as viscous dissipation and adiabatic heating and cooling (that mainly affect the temperature). However, density is still assumed constant apart from the buoyancy term, and only temperature variations drive mantle flow. For Mars, given the small pressure range compared to larger planets such as Earth and Venus and even exoplanets, the effects of compressibility are much smaller and can be neglected to first order. We thus don't believe that there are major inconsistencies between the ALMA and GAIA models.

10- The estimated downward deformation in the northern regions < 0.13 mm/year is surprisingly precise, in view of all the sources of uncertainty, even after reading the pertinent supplementary material (methods). I am also surprised to see (methods) that other possible subsidence rates (the authors quote < 0.28 mm/year) would not affect the derived viscosity structure. Why?

This is a pertinent question. Despite our efforts to thoroughly explain how we infer the viscosity structure within the given space constraints, some aspects may still be challenging to grasp. Below is a short summary explaining why a 0.13 or 0.28 mm/year deformation does not affect the viscoelastic parameters.

Our viscoelastic models predict subsidence rates varying by orders of magnitudes, ranging from several mm/year for young ice caps to 10^{-3} mm/year for older and almost fully compensated loads (Figure 3b).

Our analyses of Mars' time-variable gravity field and InSight seismic moment rate suggest that models with mm/year subsidence are inconsistent with observations. Indeed, these models are expected to measurably affect the time-variable gravity field, which is unseen, and would lead

to the nucleation of a north-polar marsquake, which has not been detected. This allows us to define a minimum age for the polar cap. The minimum loading age in turn affects our predicted present-day deformation beneath the polar cap, and together with radar constraints, these allow to infer a viscosity structure.

As shown in Figure 3b, a slight offset of the maximum subsidence rate bound from 0.13 to 0.28 mm/year would only change the minimum polar cap age from ~1.7 to ~0.9 Ma. Looking at Figure 3a and 3c, a minimum polar cap age of 0.9 Ma does not allow more models to be included. This means that our inferred mantle viscosity is unchanged.

This is now noted in line 372: *“Choosing the other value has no effect on our derived viscosity structure, but would provide a lower limit to the polar cap age of 0.9 Ma instead of 1.7 Ma.”*

11- The argument of non-observation of “marsquakes” as a support to low strains in the northern region is somewhat debatable. I would say that the InSight evidence is not in contrast with the gravity analysis, not that ‘it remains consistent’ with it.

We agree with the reviewer and have made the suggested change.

- Discussion

12- I like this section but would recommend a Conclusions section, too. To improve the presentation, the material presented here could be organized in two sections... or may be the whole section could be entitled “Discussion and Conclusions”, with the latter reported in the very last paragraph.

Thank you for the positive feedback. Unfortunately, Nature doesn’t allow sub-heading titles such as Discussion or Conclusions. We have thus changed the “Discussion” sub-heading to “Highly viscous mantle and young ice cap”, which indicates the content of the paragraph below. As noted above, we have also added sub-headings to better structure the manuscript. We hope these changes are sufficient.

- Minor points, with suggestions for improvements:

13- Figures. The figures quality is good but it is not extraordinary.

Thank you for the suggestions below, which helped improve the clarity of our Figures.

14- Figure 2. Here and in other instances one wonders what is the meaning of the labels adopted to indicate the models used. Ah, I see that this is given in the caption of Table S1. Well, why not labeling these models as (for example) M1, M2, M3, ... in Table 1 so that in Figure 2 you can just reference M1, M2, M3... It is just a suggestion, of course.

Good point. We went back and forth between a simple labelling as suggested and the current version with more information. In the revised version, we have opted for a simpler solution that doesn't display model names, but shows a colour bar indicating models with high/low crustal thickness and crustal HPE enrichment.

Detailed information on specific model names and colours is now shown in Figure S1 and Table 1.

15- Table S1. Looking at the actual values of the viscosities for the three starred models (fitting models), I would say more $\geq 10^{23}$ Pa.s than $>10^{22}$ Pa.s as reported in the abstract and in the Discussion section.

Thank you for the suggestion. We have noted a typo in our calculation for the average mantle viscosity. This parameter was estimated starting from 400 km rather than 500 km as noted in the Table. We note that this has no influence on our calculations that use the full viscosity profiles. The revised version now correctly display the average viscosity from 500 km to the core computed using a volumetric average rather than simple average.

For reference, our 3 accepted model volumetric average viscosity from 500 km to the core are **2.5e+22** (47-2.6-3.0_2 λ), **6.4e+22** (47-2.8-3.0_1.75 λ) and **4.9e+22** (47-3.0_1.25 λ) Pa s.

For clarity, the abstract now states:

"Only models with present-day high-viscosities ($2-6 \times 10^{22}$ Pa s for depth >500 km)..."

Line 140 now states:

"These models are characterised by a large present-day mantle viscosity with volumetric average values of about $2-6 \times 10^{22}$ Pa s from 500 km down to the core and thick average crust (>40 km, Figs. 3 and 4). Models with a lower viscosity would predict too much north polar deformation, while models with a higher viscosity would require crustal thicknesses larger than measured by InSight²⁶".

16- Figure S1. Ditto. See above my Figure 2 comment on the models names.

As noted in comment 14, we have removed model names from Figure 2. However, we have kept our initial model description. We believe that the current label format, which shows the InSight crustal thickness, the crust density, and the radiogenic content of the crust provides meaningful quick-to-obtain information. Because the labelling is described in the Figure caption, there is no need for the reader to go to the supplementary Table S1, which we found useful.

17- Figure S2. What is the meaning of the colors employed to draw these curves? Please add a color table or some information in the caption of the figure.

The colours are the same as in Figure 2 and Figure S1. For clarity, we have added a legend to Figure S2.

18- Figure S3. What is the ice time history adopted for the plot of the Love numbers? This is not specified and unclear. Does “Andrade” only include an elastic and a transient power law in the viscoelastic creep function, or also a Newtonian element?

The Love numbers in this plot are computed using a Heaviside step function. The Andrade rheology is defined in Melini et al (2022) [ref 8] and indeed includes a creep parameter.

We have added the following sentence to the caption to clarify this:

“Love numbers in this Figure are calculated assuming a Heaviside load history. The Andrade rheology is defined following ref⁸ and assumes an Andrade creep parameter of 0.3.”

19- Figure S5. In a) and b), please use the same y-range so to allow a fair comparison between the plots.

Thank you for the suggestion. The plots in a) and b) now use the same y-range.

20- Figures S6 and S7. The same as in Figure S5 above.

The plots in a) and b) now use the same y-range.

21- References. There are capitalization problems in a significant number of papers titles, please check. Beside this, I think that the paper is giving appropriate credit to previous work.

Good catch, this is now corrected.

I hope these comments shall be useful to improve this very good contribution.

We greatly appreciate all comments and the positive feedback. Thank you!

Referee #2:

Review of Glacial isostatic adjustment reveals Mars’ interior viscosity structure

This paper updates existing models of crustal deformation and viscosity using new input from the time-variable gravity field and the seismic moment rate determined from InSight. As past groups have done, much of this analysis relies on results from MARSIS and SHARAD plus the MOLA instrument. The addition of these new constraints allows the authors to better constrain the geophysical properties of the crust and mantle of Mars along with estimate a range of ages for the north polar ice cap.

From the north polar ice perspective, I find the results broadly consistent with established literature, and I believe that they will increase the fidelity of arguments that the north polar ice is relatively young, only a few to 10 million years old. This work may appear incremental; however, the inclusion of new datasets and constraints is quite novel. I believe this is a valuable contribution.

My expertise are not in line with the modeling work here, so I focus my review to help strengthen the arguments and support consistency, especially with citations, which are sometimes misleading, incorrect, or out of date. I have no major comments, but there two moderate comments and several opportunities for minor modifications that can strengthen the manuscript.

We greatly appreciate the positive feedback from reviewer Isaac Smith. All comments are addressed in detail below.

Two moderate comments are that 1) the authors provide ranges but not best fits. Obviously, this type of analysis inherently comes with error bars, and it is important to show them or provide valid ranges that fit the given data. That said, it would be useful to the community of researchers who will read this paper to know the best fit, given the new inputs to the model.

This is an interesting point. Our analysis combines multiple datasets and methods, with each having their own limitations and uncertainties. Because of this, the derived range of plausible solution from this study cannot be treated as a Gaussian with a best-fit and standard deviation. From a radar perspective, our radar measurements favour models predicting the least amount of deformation. This is achieved by having a very young ice cap (1.7 Ma), with the lightest mixtures of ices and dust, and highest mantle viscosity and crustal thickness (model 47-3.0_1.25 λ). However, we do not believe this particular solution to be better/preferable (in a statistical sense) compared to others cited throughout the manuscript. For that reason, we prefer to report the full plausible range of solutions given the uncertainty inherent to each approach.

And 2) The manuscript does not compare the new results to past results. It kind of pretends that past work of this type does not exist, and so readers are left wondering how different is this than the Phillips 2008 results.

Thank you for raising this important point. We do not agree with the reviewer here. Previous work is mentioned line 40 [the work of Phillips is ref 5]:

“Most previous analyses have assumed the polar deformation to be at equilibrium, which is only valid if the time elapsed since the polar cap formed is greater than the time required for viscous adjustments^{5,6}. Geologic observations and global climate models suggest the north polar cap is only a few million years old, but the exact age remains uncertain^{22,23}. Due to this young age, calculations suggested that viscoelastic relaxation could result in estimating thinner elastic lithospheres and higher heat flows beneath the north polar cap^{5,6}. However, these models assumed a constant mantle viscosity, did not account for the ice loading history, and only

considered a single wavelength to represent the load. As a result, limited insights into the possible effects of viscoelastic relaxation were provided.”

We believe the above text to adequately presents earlier work and their limitations.

As for a comparison between our work and previous studies, we note that to date, there has been no detailed study of viscoelastic relaxation for the Martian polar caps attempting to constrain both the age of the polar cap and Mars’ mantle viscosity. To our knowledge, here are some relevant papers mentioning the viscoelastic response of the interior and how they differ from our work:

- Greve et al. (2003) [cited in the manuscript] constrained the lithosphere thickness but assumed a relaxed mantle, did not have many measurable constraints available at the time (i.e., radar data, gravity, seismic data) and was thus unable to provide insights on the interior viscosity and polar cap age.
- Phillips et al. (2008) [also Broquet et al., 2020, both cited] focused on the elastic response of the interior to the polar cap load. This approach assumes a mantle at isostatic equilibrium, which is obviously not appropriate for a short-term loading process on present-day Mars. In the supplementary materials, of both papers the authors briefly considered the effects of viscous relaxation. However, in addition to not using seismic data (not available at the time) nor gravity data, these studies used an overly simplistic parametrization of that process assuming that:
 - 1) The polar cap formed instantly
 - 2) The viscosity of the mantle has a single and constant value
 - 3) The viscoelastic response can be described by a single wavelength (spherical harmonic degree 8). Absolute viscoelastic deformation were not computed and compared to radar data. Rather the behaviour of a single time-dependent Love number was discussed.
- Ojha et al. (2019) [cited] used a finite-element approach to estimate a crustal/mantle heat flow. That work did not infer the viscosity of the interior, nor did it constrain the ice loading history or polar cap age. No seismic (not available at the time) or gravity data was used.

While our goal is not to undermine the important work developed in these studies, we strongly believe that previous work did not adequately address the viscoelastic response of the interior to the north polar cap. The brief mentions to the transient (viscoelastic) response of the interior to the polar cap loading cannot be considered as a detailed prior investigation of this process that can be compared to our work.

Nevertheless, we agree with the reviewer’s comment below that our reference to prior work as having 'neglected' viscoelasticity was inappropriate. In the revised manuscript, this was changed to:

“This effect would prominently affect all estimates of the strength of the elastic lithosphere and planetary thermal state, where viscous relaxation is not addressed adequately^{5,6,35”}

We hope the reviewer will find our response to this point satisfactory.

-Isaac Smith

Unfortunately, there are no line numbers, so I have to quote directly from the text

We apologize for this issue, which is now corrected.

Paragraph 3, “ongoing plume activity (Fig S1).”

-Figure S1 does not mention plume activity, so some change should be made here

Thank you for catching this. We have now moved the Fig. S1 reference to earlier in the sentence: *“geodynamic evolution models^{7,17,18} struggle to explain the thick and cold lithosphere inferred at the north pole (Fig. S1)”*

“suggest that the current northern ice cap formed over the past 4 Ma, with ice-sheet thickness increasing linearly in response to a gradual decline in polar insolation^{23”}.

—Reference 23 does not provide a linear thickness increase. It is more sawtooth. Greve et al 2006 modeled the same sawtooth increase in thickness, and Smith et al., 2016 demonstrated that the stratigraphy includes unconformities that represent periods of material loss between accumulation. Please rephrase. This issue shows up again in the following paragraph

The reviewer is correct and we have removed “linearly” in that sentence. We have added a citation to Smith et al. 2016 line 89 [ref 36]:

“Although non-linearities in the ice accumulation history are expected³⁶, these would have a negligible effect on present-day deformations (Fig. S3)”

We have also added a supplementary Figure (Fig. S3) showing that the ice accumulation history does not significantly affect the recent / final deformation considered in our analyses.

“Because the age and long-term loading history of the northern ice cap are poorly known, we assume a linear increase in ice-sheet thickness over time, leading to its current state”

With the new added Figure, we believe this assumption to be adequate. Any other assumption would require an additional parameter, such as the sawtooth function characteristic width.

"This effect would prominently affect all estimates of the strength of the elastic lithosphere and planetary thermal state that have neglected viscous relaxation^{5, 6}."

-this is misleading. In the introduction for Phillips 2008, they said "Alternatively, the response to the load may be in a transient state controlled by mantle viscosity"

Thank you for raising that point, which we fully agree with. We have changed "*neglected*" to "*not adequately addressed*".

"surface age (100 kyr, ref35)"

-this is an out of date surface age. A much more recent paper suggests 1.5 kyr for the surface age: Landis 2016

We thank the reviewer for the more up-to-date reference. In an effort to save text space, we have decided to fully remove this statement and the associated reference.

"We note that changing the ice accumulation history from linear to exponential or sinusoidal in our viscoelastic calculations has a negligible effect compared to the uncertainty in the mantle viscosity structure."

-I think you are probably right, but this is not established in the manuscript. Perhaps you could add more in supplemental information. Readers will wish to see how these changes affect the outcome, especially given that ice accumulation has not been linear.

Thank you for the great suggestion. We have added a supplementary Figure (S3) showing that the ice accumulation history (linear, sinusoidal, sawtooth) have little effect on the present-day deformation.

"Changes in the seasonal cap whose load is 3 orders of magnitude lower than the perennial cap"

-The authors are ignoring past episodes of atmospheric collapse that could have added much more CO₂ mass load. Please see Buhler et al., 2019, 2021 or Bierson 2016 and Manning 2016

As noted above, the specific shape of the ice accumulation function cannot be constrained and has little influence on our analyses. In addition, the above references discuss CO2 deposition on the southern polar cap, a process that would likely differ at the north.

As noted above, we have also added a citation to Smith et al 2016 [ref 36] (line 89):
"Although non-linearities in the ice accumulation history are expected³⁶, these would have a negligible effect on present-day deformations (Fig. S3)"

"The present-day north polar cap has a thickness of about 3.1 km and bulk density of about $1200 \pm 300 \text{ kg m}^{-3}$, as constrained from radar analyses⁶, 34"

-34 doesn't mention density. Is this a correct citation?

Reference 6 does constrain both the density and thickness of the polar deposits. Reference 34 provides some insights on the polar cap thickness, which we believe to be appropriate.

"Because of its higher elevation, the south polar cap is expected to only accumulate small amount of dry ice³⁹, which are here neglected"

-there are newer and better references for this statement, e.g. Thomas, P.C., Calvin, W., Cantor, B., Haberle, R., James, P.B., Lee, S.W., 2016. Mass balance of Mars' residual south polar cap from CTX images and other data. *Icarus* 268, 118–130. <https://doi.org/10.1016/j.icarus.2015.12.038>

Thank you, we have updated the reference as suggested.

"If the northern strain rates were too large, crustal failure and faulting would occur, potentially leading to the detection of a marsquake by InSight"

-maybe this is the better argument - that faults are not observed. Can the authors say anything about the size of expected faults if the crust were to fail brittlely?

Good point, but there may be a slight misconception here. A quake / crustal failure is not necessarily associated with a fault outcropping on the surface. There are indeed only a handful of faults in the northern regions, most of which are compressive features (wrinkle ridges). However, we don't think that the lack of observable fault can be used as a constraint. In addition, the size of a visible fault would likely be the result of multiple marsquakes over a long time rather than being indicative for the size of the most recent events.

"Based on predicted magnitudes from the InSight catalogue, a 3.8 to 3.9 magnitude event originating from the polar regions (75–90°N) could be detectable¹⁰. Thus, the non-observation of such marsquakes during the InSight mission can be used"

-tricky argument given that InSight was not active for long. Can the authors say statically how many magnitude 3.9 earthquakes would be expected during the lifetime of InSight? Without that, this argument is meaningless. Say, we only expect one every 5 years in the north polar region, and InSight missed it, would that tell us anything?

Thank you for the comment. The InSight catalogue allowed us to determine the detectability of marsquakes by the SEIS instrument. The analysis of Knapmeyer et al (2023) [cited] showed that a magnitude >3.8 event in the northern regions (75-90°N) *could* be detectable by SEIS, if it occurred. In the magnitude-frequency distribution Fig. 2b of Knapmeyer [REDACTED] one finds that we had 4 events with a magnitude exceeding 3.8 globally. One also finds that 10 marsquakes with a smaller magnitude have been detected from distances even greater than the northern regions. Together, these indicate that high-magnitude quakes and distant events have been detected.

Using the above information, we turn to the Kostrov equation and convert the seismic moment of a magnitude 3.8 quake to a geodetic moment to infer a maximum strain rate. This gives us a strain rate threshold above which the crust would brittlely fail and induce a SEIS-detectable 3.8 magnitude quake. Given that we have not observed a 3.8 magnitude quake in the northern region in the lifespan of InSight, we can say that the northern strain rate during that period must have been less than a given value, which in turns allows us to limit our parameter space.

We do agree that strain could be accumulating at depth rather than being actively released and nucleating marsquakes (this even on Earth is impossible to constrain). This effect makes the comparison of a long-term deformation process to 2 years of seismic measurements subject to uncertainties. However, and as noted in the text, the InSight moment rate provides a weaker constraint on the polar cap age and is thus not used to delimit our parameter space. Still, this seismic moment analysis, which is based on extensive work linking geodetic deformation and seismic measurements on Earth, suggests low polar strain rates. These are remarkably consistent with our gravity measurements and the seismic moment analysis is thus deemed relevant to highlight in the paper.

[REDACTED]

"One way to help reduce the ongoing deformation, and thus our inferred interior viscosity, would require the present-day ice cap to have formed on the sedimentary-infilled flexural trough of a former ice cap. In that framework, the ongoing downward deformation would compete with the rebound related to the past ice-sheet. However, while such sedimentary infilling may be related to the basal unit beneath the central portion of the polar layered deposits³⁴, it is not seen beneath the Gemina Lingula region near 80°N (Fig. 1). Therefore, this competing effect alone cannot account for the negligible flexure observed in that region."

-I like that you brought this up. Would it be possible to assume that the basal unit (which Nerozzi found the volume for) is 1 Ga old (Tanaka 2005?) and rerun these models? I think the distribution of basal unit is publicly available. I would be curious about how much this lowers the upper age limit. Please state that here.

That's a great question. We initially ran some models with this scenario, but decided not to display them. The work of Nerozzi has shown that the basal unit (BU) has a volume of about 25% of that of the entire north polar cap [NPLD + BU]. Having a 1 Ga BU would lead to some more deformation, because the BU-induced flexure would have had a longer time to develop.

Below, we show a calculation similar to that presented in Figure S6, S7, S8 displaying how the polar cap thickness and associated flexure vary (here only looking at the maximum value). We compare a model considering an early formed BU at 1 Ga and a NPLD forming linearly over 8 Ma to a model considering a complete polar cap (NPLD + BU) formation at 8 Ma.

Figure: Maximum deformation beneath the polar cap as a function of time, before present, (top) considering different accumulation histories (bottom) for high (left) and low (right) mantle viscosity models. The dashed line considers a formation at 8 Ma for the NPLD + BU, whereas the solid lines consider a BU formation at 1 Ga and a NPLD formation at 8 Ma.

This plot shows that considering an older BU moderately increases (few tens of meters) the present-day deformation. Accounting for this would lead to our model predicting a slightly younger maximum age for the polar cap, which we estimate to be ~5-8 Ma. Assuming a linear increase in the BU thickness, however, would make an updated solution even closer to our initial model with a maximum age close to 12 Ma reported in the main text.

"MARSIS radar analyses were performed at 213 locations spatially scattered across the polar deposits.

All available MARSIS radargrams close to each location were investigated, the reflections arising from the icy surface and the ice/substratum interface were visually identified. Using the same framework, analyses of SHARAD radargrams were performed at 78 locations."

-I don't think Figure 1b shows 300 locations. Please clarify in the text or figure caption.

Thank you for catching this. Indeed, the SHARAD locations overlap with MARSIS. We have updated the above sentence as:

“Using the same framework, analyses of SHARAD radargrams were performed at those 213 locations. Because of its higher frequency, SHARAD does not penetrate through the sand-rich basal unit and we here only retain locations where the ice/crust interface is observed (N=78).”

"The radar thickness depends on the dielectric properties of the icy materials, which are unknown."

-This is a rather misleading statement. Grima et al., 2009 completed an excellent analysis that the authors are ignoring here. That paper is a critical data point that has been cited hundreds of times. Ignoring it here does allow the authors to invent a number to use for ϵ' , but that number is not realistic. I understand that their analysis does allow values from 2.5 to 3.5, and that range may have value for their model, but as an avid researcher of north polar ice, I am very interested to hear what choosing a value of 3.1 or 3.15 would do to the results.

We are aware of the interesting work presented in Grima et al. Unfortunately, due to word space and citation limitations, we were not able to reference that work.

In addition, we would like to emphasize that work of Grima et al. only investigated a restricted portion of the north polar cap, meaning that ice–dust mixtures with different dielectric properties may exist elsewhere. Given that we are interested in the bulk polar cap, we have preferred to explore a range of possible mixtures. In addition, Grima et al. did not consider any potential lithosphere flexure beneath the polar cap, which would affect the retrieved topography-derived ice thickness and inferred dielectric constants (making them lower). While the no-flexure assumption may be correct, there is a bit of a circular argument in saying that the dielectric constant is 3.1 because there is no flexure and saying that there is no flexure because the dielectric constant is 3.1.

Rather than fixing one of the other, our inversion explores all possible combinations that are then restricted based on plausible mixtures and bulk density constraints. For that reason, we don't agree that we "invent" a number for ϵ' , as our work covers a reasonable range for ice and dust mixtures.

To better reflect this, we have changed the mentioned statement as (line 216):

“The radar thickness depends on the dielectric properties of the icy materials, which are not uniquely known for the entire north polar cap.”

We do acknowledge the interest in knowing what the results would be with a 3.1 dielectric constant. If we restrict our inversion to ice mixtures with a minimum dielectric constant of 3.1, we obtain a maximum polar cap age of 2 Ma.

Same with the following a little lower in the text

"For context Fig. 1b shows a flexure map beneath the northern cap that was obtained by assuming a dielectric constant of 3.0 for our MARSIS and SHARAD radar measurements."

The work of Grima indicates a dielectric constant of 3.10 ± 0.12 (2.98 to 3.22), which is consistent with our example plot assuming a dielectric constant of 3.0.

In the section about past ice caps

"This model starts at 10 Ma with a past northern ice sheet that disappears at about 8 Ma. After a time-interval with no northern ice cap, a new northern ice sheet forms with a linear increase in thickness from 4 Ma to present-day (Fig. S6). A key outstanding unknown is what was the state of the 10 Ma ice sheet and whether it was long-standing and at equilibrium or more recently formed. In one model, we assume that the previous ice cap was present over the last 500 Ma and thus at equilibrium (Fig. S6) and in a second model, we assume that it formed at 14 Ma (Fig. S7). The state of this past ice sheet drastically affects its influence on the present-day deformation. In the case where the ice cap was long-lasting, the present-day deformation is increased by up to 70 m, whereas"

-Why does the analysis ignore the basal unit that has a significant volume and mass over 10^9 years?

This analysis was developed to show that the existence of long-standing ice caps or the removal of past ice caps has little influence on the present-day flexure.

As noted above, considering an older basal unit would slightly affect our derived ages for the polar cap.

Figure 1

Panel B: how many points are included?

We have updated the caption as:

"interpolated deformed basement assuming a real dielectric constant of 3.0 based on 78 SHARAD and 213 MARSIS radar measurements (b, Methods)"

As noted above, we have also added the following statement in the Methods section:

"Using the same framework, analyses of SHARAD radargrams were performed at those 213 locations. Because of its higher frequency, SHARAD does not penetrate through the sand-rich basal unit and we here only retain locations where the ice/crust interface is observed (N=78)."

How was panel c made? There is no attribution, although I expect it was pulled from a past publication. Also, 3.0 is inconsistent with Grima 2009.

The plot was made by loading the orbit 51924 SHARAD radargram, detecting the surface based on the brightest echo (with some smoothing to avoid picking a bright subsurface echo), and applying a depth-correction to all points beneath the surface assuming a dielectric constant of 3. This is now noted in the Figure caption:

“The radargram apparent depth was corrected by assigning real permittivities of 1.0 and 3.0 above and below the detected ground surface”

As noted above, the work of Grima indicates a dielectric constant of 3.10 ± 0.12 , which is consistent with our example plot assuming a dielectric constant of 3.0.

And, to support the argument here about deflection, there may be value in extending the radar image further beyond the layered ice (south in both directions) to demonstrate that the lower boundary is coplanar to the surrounding terrain

Very good point, thanks. We have now slightly extended the Figure towards point B where the crust is visible (Point A is covered by Olympia Planum).

Figure S4

The color scale here is hard to see, please choose different colors. I have no difficulty distinguishing colors, but people with vision impairment will not be able to tell the difference between blue with black border and blue with purple border. Same with yellow. This really needs to be changed to distinguish between the planetary average and the polar region. Panel b does not have a color legend.

Thank you for catching this. We have now modified that Figure (now Figure S5) to show two slightly different colours to delimit the global and northern region averages.